# Formalizing the Generalization-Forgetting Trade-Off in Continual Learning

**R. Krishnan**[1] **and Prasanna Balaprakash**[1,2]
[1]Mathematics and Computer Science Division
[2]Leadership Computing Facility
Argonne National Laboratory
*kraghavan,pbalapra@anl.gov*

## Abstract

We formulate the continual learning problem via dynamic programming and model the trade-off between catastrophic forgetting and generalization as a two-player sequential game. In this approach, player 1 maximizes the cost due to lack of generalization whereas player 2 minimizes the cost due to increased catastrophic forgetting. We show theoretically and experimentally that a balance point between the two players exists for each task and that this point is stable (once the balance is achieved, the two players stay at the balance point). Next, we introduce balanced continual learning (BCL), which is designed to attain balance between generalization and forgetting, and we empirically demonstrate that BCL is comparable to or better than the state of the art.

## 1 Introduction

In continual learning (CL), we incrementally adapt a model to learn tasks (defined according to the problem at hand) observed sequentially. CL has two main objectives: maintain long-term memory (remember previous tasks) and navigate new experiences continually (quickly adapt to new tasks). An important characterization of these objectives is provided by the stability-plasticity dilemma [9], where the primary challenge is to balance network stability (preserve past knowledge; minimize catastrophic forgetting) and plasticity (rapidly learn from new experiences; generalize quickly). This balance provides a natural objective for CL: *balance forgetting and generalization.*

Traditional CL methods either minimize catastrophic forgetting or improve quick generalization but do not model both. For example, common solutions to the catastrophic forgetting issue include (1) representation-driven approaches [49, 25], (2) regularization approaches [27, 2, 34, 16, 48, 48, 24, 37, 10, 43], and (3) memory/experience replay [31, 32, 11, 12, 17]. Solutions to the generalization problem include representation-learning approaches (matching nets [45], prototypical networks [42], and metalearning approaches [18, 19, 8, 47]). More recently, several approaches [35, 16, 46, 23, 48, 15] have been introduced that combine methods designed for quick generalization with frameworks designed to minimize forgetting.

The aforementioned CL approaches naively minimize a loss function (combination of forgetting and generalization loss) but do not explicitly account for the trade-off in their optimization setup. The first work to formalize this trade-off was presented in meta-experience replay (MER) [38], where the forgetting-generalization trade-off was posed as a gradient alignment problem. Although MER provides a promising methodology for CL, the balance between forgetting and generalization is enforced with several hyperparameters. Therefore, two key challenges arise: (1) lack of theoretical tools that study the existence (*under what conditions does a balance point between generalization and forgetting exists?*) and stability (*can this balance be realistically achieved?*) of a balance point

35th Conference on Neural Information Processing Systems (NeurIPS 2021).

and (2) lack of a systematic approach to achieve the balance point. We address these challenges in this paper.

We describe a framework where we first formulate CL as a sequential decision-making problem and seek to minimize a cost function summed over the complete lifetime of the model. At any time $k$, given that the future tasks are not available, the calculation of the cost function becomes intractable. To circumvent this issue, we use Bellman's principle of optimality [4] and recast the CL problem to model the catastrophic forgetting cost on the previous tasks and generalization cost on the new task. We show that equivalent performance on an infinite number of tasks is not practical (Lemma 1 and Corollary 1) and that tasks must be prioritized. To achieve a balance between forgetting and

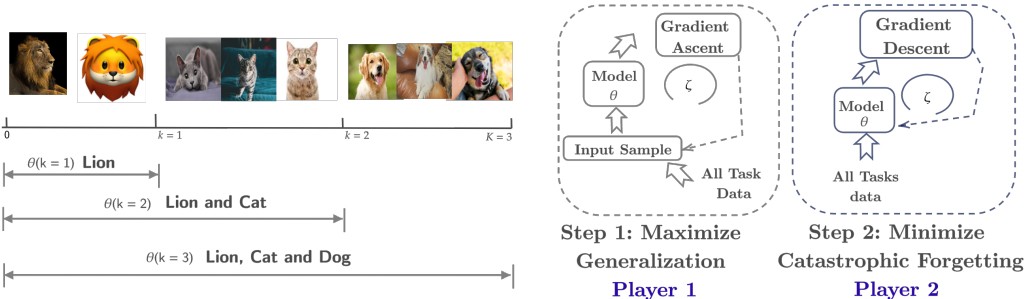

Figure 1: (left) Exemplary CL problem: the lifetime of the model can be split into three intervals. At $k = 1$ we seek to recognize lions; at $k = 2$ we seek to recognize both lions and cats; and at $k = 3$ we seek to recognize cats, lions, and dogs. (right) Illustration of the proposed method: our methodology comprises an interplay between two players. The first player maximizes generalization by simulating maximum discrepancy between two tasks. The second player minimizes forgetting by adapting to maximum discrepancy .

generalization, we pose the trade-off as a saddle point problem where we designate one player for maximizing the generalization cost (player 1) and another for minimizing the forgetting cost (player 2). We prove mathematically that there exists at least one saddle point between generalization and forgetting for each new task (Theorem 1). Furthermore, we show that this saddle point can be attained asymptotically (Theorem 2) when player strategies are chosen as gradient ascent-descent. We then introduce balanced continual learning (BCL), a new algorithm to achieve this saddle point. In our algorithm (see Fig. 1 for a description of BCL), the generalization cost is computed by training and evaluating the model on given new task data. The catastrophic forgetting cost is computed by evaluating the model on the task memory (previous tasks). We first maximize the generalization cost and then minimize the catastrophic forgetting cost to achieve the balance. We compare our approach with other methods such as elastic weight consolidation (EWC) [27], online EWC [40], and MER [38] on continual learning benchmark data sets [21] to show that BCL is better than or comparable to the state-of-the-art methods. Moreover, we also show in simulation that our theoretical framework is appropriate for understanding the continual learning problem. The contributions of this paper are (1) a theoretical framework to study the CL problem, (2) BCL, a method to attain balance between forgetting and generalization, and (3) advancement of the state of the art in CL.

## 2  Problem Formulation

We use $\mathbb{R}$ to denote the set of real numbers and $\mathbb{N}$ to denote the set of natural numbers. We use $\|.\|$ to denote the Euclidean norm for vectors and the Frobenius norm for matrices, while using bold symbols to illustrate matrices and vectors. We define an interval $[0, K), K \in \mathbb{N}$ and let $p(\mathcal{Q})$ be the distribution over all the tasks observed in this interval. For any $k \in [0, K)$, we define a parametric model $g(.)$ with $\boldsymbol{y}_k = g(\boldsymbol{x}_k; \boldsymbol{\theta}_k)$, where $\boldsymbol{\theta}_k$ is a vector comprising all parameters of the model with $\boldsymbol{x}_k \in \mathcal{X}_k$. Let $n$ be the number of samples and $m$ be the number of dimensions. Suppose a task at $k | k \in [0, K)$ is observed and denoted as $\mathcal{Q}_k : \mathcal{Q}_k \sim p(\mathcal{Q})$, where $\mathcal{Q}_k = \{\mathcal{X}_k, \ell_k\}$ is a tuple with $\mathcal{X}_k \in \mathbb{R}^{n \times m}$ being the input data and $\ell_k$ quantifies the loss incurred by $\mathcal{X}_k$ using the model $g$ for the task at $k$. We denote a sequence of $\boldsymbol{\theta}_k$ as $\boldsymbol{u}_{k:K} = \{\boldsymbol{\theta}_\tau \in \Omega_\theta, k \leq \tau \leq K\}$, with $\Omega_\theta$ being the compact (feasible) set for the parameters. We denote the optimal value with a superscript $(*)$; for instance, we use $\boldsymbol{\theta}_k^{(*)}$ to denote the optimal value of $\boldsymbol{\theta}_k$ at task $k$. In this paper we use balance

point, equilibrium point, and saddle point to refer to the point of balance between generalization and forgetting. We interchange between these terms whenever convenient for the discussion. We will use $\nabla_{(j)} i$ to denote the gradient of $i$ with respect to $j$ and $\Delta i$ to denote the first difference in discrete time.

An exemplary CL problem is described in Fig. 1 where we address a total of $K = 3$ tasks. To particularize the idea in Fig. 1, we define the cost (combination of catastrophic cost and generalization cost) at any instant $k$ as $J_k(\boldsymbol{\theta}_k) = \gamma_k \ell_k + \sum_{\tau=0}^{k-1} \gamma_\tau \ell_\tau$, where $\ell_\tau$ is computed on task $\mathcal{Q}_\tau$ with $\gamma_\tau$ describing the contribution of $\mathcal{Q}_\tau$ to this sum.

To solve the problem at $k$, we seek $\boldsymbol{\theta}_k$ to minimize $J_k(\boldsymbol{\theta}_k)$. Similarly, to solve the problem in the complete interval $[0, K]$, we seek a $\boldsymbol{\theta}_k$ to minimize $J_k(\boldsymbol{\theta}_k)$ for each $k \in [0, K]$. In other words we seek to obtain $\boldsymbol{\theta}_k$ for each task such that the cost $J_k(\boldsymbol{\theta}_k)$ is minimized. Therefore, the optimization problem for the overall CL problem (overarching goal of CL) is provided as the minimization of the cumulative cost $V_k(\boldsymbol{u}_{k:K}) = \sum_{\tau=k}^{K} \beta_\tau J_\tau(\boldsymbol{\theta}_\tau)$ such that $V_k^{(*)}$, is given as

$$V_k^{(*)} = min_{\boldsymbol{u}_{k:K}} V_k(\boldsymbol{u}_{k:K}), \tag{1}$$

with $0 \le \beta_\tau \le 1$ being the contribution of $J_\tau$ and $\boldsymbol{u}_{k:K}$ being a weight sequence of length $K - k$.

Within this formulation, two parameters determine the contributions of tasks: $\gamma_\tau$, the contribution of each task in the past, and $\beta_\tau$, the contribution of tasks in the future. To successfully solve the optimization problem, $V_k(\boldsymbol{u}_{k:K})$ must be bounded and differentiable, typically ensured by the choice of $\gamma_\tau, \beta_\tau$. Lemma 1 (full statement and proof in Appendix A) states that *equivalent performance cannot be guaranteed for an infinite number of tasks*. Furthermore, Corollary 1 (full statement and proof in Appendix A) demonstrates that *if the task contributions are prioritized, the differentiability and boundedness of $J_\tau(\boldsymbol{\theta}_\tau)$ can be ensured*. A similar result was proved in [28], where a CL problem with infinite memory was shown to be NP-hard from a set theoretic perspective. These results (both ours and in [28]) demonstrate that a CL methodology cannot provide perfect performance on a large number of tasks and that tasks must be prioritized.

Despite these invaluable insights, the data corresponding to future tasks (interval $[k, K]$) is not available, and therefore $V_k(\boldsymbol{u}_{k:K})$ cannot be evaluated. The optimization problem in Eq. (1) naively minimizes the cost (due to both previous tasks and new tasks) and does not provide any explicit modeling of the trade-off between forgetting and generalization. Furthermore, $\boldsymbol{u}_{k:K}$, the solution to Eq. (1) is a sequence of parameters, and it is not feasible to maintain $\boldsymbol{u}_{k:K}$ for a large number of tasks. Because of these three issues, the problem is theoretically intractable in its current form.

We will first recast the problem using tools from dynamic programming [29], specifically Bellman's principle of optimality, and derive a difference equation that summarizes the complete dynamics for the CL problem. Then, we will formulate a two-player differential game where we seek a saddle point solution to balance generalization and forgetting.

## 3 Dynamics of Continual Learning

Let $V_k^{(*)} = min_{\boldsymbol{u}_{k:K}} \sum_{\tau=k}^{K} \beta_\tau J_\tau(\boldsymbol{\theta}_\tau)$; the dynamics of CL (the behavior of optimal cost with respect to $k$) is provided as

$$\Delta V_k^{(*)} = -min_{\boldsymbol{\theta}_k \in \Omega_\theta} \big[ \beta_k J_k(\boldsymbol{\theta}_k) + \big( \langle \nabla_{\boldsymbol{\theta}_k} V_k^{(*)}, \Delta \boldsymbol{\theta}_k \rangle + \langle \nabla_{\boldsymbol{x}_k} V_k^{(*)}, \Delta \boldsymbol{x}_k \rangle \big) \big]. \tag{2}$$

The derivation is presented in Appendix A (refer to Proposition 1). Note that $V_k^{(*)}$ is the minima for the overarching CL problem in Eq. (2) and $\Delta V_k^{(*)}$ represents the change in $V_k^{(*)}$ upon introduction of a new task (we hitherto refer to this as perturbations). Zero perturbations ($\Delta V_k^{(*)} = 0$) implies that the introduction of a new task does not impact our current solution; that is, the optimal solution on all previous tasks is optimal on the new task as well. Therefore, the smaller the perturbations, the better the performance of a model on all tasks, thus providing our main objective: minimize the perturbations ($\Delta V_k^{(*)}$). In Eq. 2, $\Delta V_k^{(*)}$ is quantified by three terms: the cost contribution from all the previous tasks and the new task $J_k(\boldsymbol{\theta}_k)$; the change in the optimal cost due to the change in the parameters $\langle \nabla_{\boldsymbol{\theta}_k} V_k^{(*)}, \Delta \boldsymbol{\theta}_k \rangle$; and the change in the optimal cost due to the change in the input (introduction of new task) $\langle \nabla_{\boldsymbol{x}_k} V_k^{(*)}, \Delta \boldsymbol{x}_k \rangle$.

The first issue with the cumulative CL problem (Eq. (1)) can be attributed to the need for information from the future. In Eq. (2), all information from the future is approximated by using the data from the new and the previous tasks. Therefore, the solution of the CL problem can directly be obtained by solving Eq. (2) using all the available data. Thus, $min_{\boldsymbol{\theta}_k \in \Omega}\big[H(\Delta \boldsymbol{x}_k, \boldsymbol{\theta}_k)\big]$ yields $\Delta V_k^{(*)} \approx 0$ for $\beta > 0$, with $H(\Delta \boldsymbol{x}_k, \boldsymbol{\theta}_k) = \beta_k J_k(\boldsymbol{\theta}_k) + \langle \nabla_{\boldsymbol{\theta}_k} V_k^{(*)}, \Delta \boldsymbol{\theta}_k \rangle + \langle \nabla_{\boldsymbol{x}_k} V_k^{(*)}, \Delta \boldsymbol{x}_k \rangle$. Essentially, minimizing $H(\Delta \boldsymbol{x}_k, \boldsymbol{\theta}_k)$ would minimize the perturbations introduced by any new task $k$.

In Eq. (2), the first and the third term quantify generalization and the second term quantifies forgetting. A model exhibits generalization when it successfully adapts to a new task (minimizes the first and the third term in Eq. (2)). The degree of generalization depends on the discrepancy between the previous tasks and the new task (numerical value of the third term in Eq. (2)) and the worst-case discrepancy prompts maximum generalization. Quantification of generalization is provided by $\Delta \boldsymbol{x}_k$ that summarizes the discrepancy between subsequent tasks. However, $\Delta \boldsymbol{x}_k = \boldsymbol{x}_{k+1} - \boldsymbol{x}_k$, and $\boldsymbol{x}_{k+1}$ is unknown at $k$. Therefore, we simulate worst-case discrepancy by iteratively updating $\Delta \boldsymbol{x}_k$ through gradient ascent in order to maximize $H(\Delta \boldsymbol{x}_k, \boldsymbol{\theta}_k)$; thus maximizing generalization. However, large discrepancy increases forgetting, and worst-case discrepancy yields maximum forgetting. Therefore, once maximum generalization is simulated, minimizing forgetting (update $\boldsymbol{\theta}_k$ by gradient descent) under maximum generalization provides the balance.

To formalize our idea, let us indicate the iteration index at $k$ by $i$ and write $\Delta \boldsymbol{x}_k$ as $\Delta \boldsymbol{x}_k^{(i)}$ and $\boldsymbol{\theta}_k$ as $\boldsymbol{\theta}_k^{(i)}$ with $H(\Delta \boldsymbol{x}_k, \boldsymbol{\theta}_k)$ as $H(\Delta \boldsymbol{x}_k^{(i)}, \boldsymbol{\theta}_k^{(i)})$ (for simplicity of notation, we will denote $H(\Delta \boldsymbol{x}_k^{(i)}, \boldsymbol{\theta}_k^{(i)})$ as $H$ whenever convenient). Next, we write

$$
\min_{\boldsymbol{\theta}_k^{(i)} \in \Omega_\theta} \left[ H(\Delta \boldsymbol{x}_k^{(i)}, \boldsymbol{\theta}_k^{(i)}) \right] = \min_{\boldsymbol{\theta}_k^{(i)} \in \Omega_\theta} \left[ \beta_k J_k(\boldsymbol{\theta}_k^{(i)}) + \langle \nabla_{\boldsymbol{\theta}_k^{(i)}} V_k^{(*)}, \Delta \boldsymbol{\theta}_k^{(i)} \rangle + \langle \nabla_{\boldsymbol{x}_k^{(i)}} V_k^{(*)}, \Delta \boldsymbol{x}_k^{(i)} \rangle \right]
$$
$$
\leq \min_{\boldsymbol{\theta}_k^{(i)} \in \Omega_\theta} \left[ \beta_k J_k(\boldsymbol{\theta}_k^{(i)}) + \langle \nabla_{\boldsymbol{\theta}_k^{(i)}} V_k^{(*)}, \Delta \boldsymbol{\theta}_k^{(i)} \rangle + \max_{\Delta \boldsymbol{x}_k^{(i)} \sim p(\boldsymbol{\mathcal{Q}})} \langle \nabla_{\boldsymbol{x}_k^{(i)}} V_k^{(*)}, \Delta \boldsymbol{x}_k^{(i)} \rangle \right]
$$
$$
\leq \min_{\boldsymbol{\theta}_k^{(i)} \in \Omega_\theta} \max_{\Delta \boldsymbol{x}_k^{(i)} \sim p(\boldsymbol{\mathcal{Q}})} \left[ H(\Delta \boldsymbol{x}_k^{(i)}, \boldsymbol{\theta}_k^{(i)}) \right].
$$

(3)

In Eq. (3), we seek the solution pair $(\Delta \boldsymbol{x}_k^{(*)}, \boldsymbol{\theta}_k^{(*)}) \in (\Omega_\theta, \Omega_{\Delta \boldsymbol{x}_k^{(*)}})$, where $\Delta \boldsymbol{x}_k^{(*)}$ maximizes $H$ (maximizing player, player 1) while $\boldsymbol{\theta}_k^{(*)}$ minimizes $H$ (minimizing player, player 2) where $(\Omega_\theta, \Omega_{\Delta \boldsymbol{x}_k^{(*)}})$ are the feasible sets for $\Delta \boldsymbol{x}_k^{(i)}$ and $\boldsymbol{\theta}_k^{(i)}$ respectively. The solution is attained, and $(\Delta \boldsymbol{x}_k^{(*)}, \boldsymbol{\theta}_k^{(*)})$ is said to be the equilibrium point when it satisfies the following condition:

$$
H(\Delta \boldsymbol{x}_k^{(*)}, \boldsymbol{\theta}_k^{(i)}) \geq H(\Delta \boldsymbol{x}_k^{(*)}, \boldsymbol{\theta}_k^{(*)}) \geq H(\Delta \boldsymbol{x}_k^{(i)}, \boldsymbol{\theta}_k^{(*)}). \tag{4}
$$

### 3.1 Theoretical Analysis

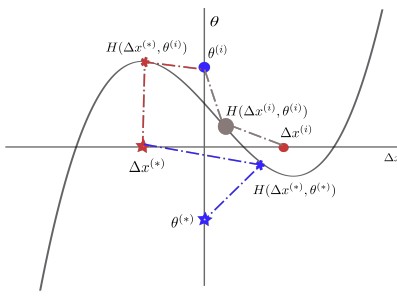

Figure 2: Illustration of the proofs. $\Delta \boldsymbol{x}$ (player 1) is the horizontal axis, and the vertical axis indicates $\boldsymbol{\theta}$ (player 2) where the curve indicates H. If we start from the red circle for player 1 (player 2 is fixed at the blue circle), H is increasing (goes from a grey circle to a red asterisk) with player 1 reaching the red asterisk. Next, start from the blue circle ($\boldsymbol{\theta}$ is at the red asterisk), the cost decreases.

With our formulation, two key questions arise: Does our problem setup have an equilibrium point satisfying Eq. (4)? and how can one attain this equilibrium point? We answer these questions with Theorems 1 and 2, respectively. Full statements and proofs are provided in Appendix A.

To illustrate the theory, we refer to Fig. 2, where the initial values for the two players are characterized by the pair $\{\boldsymbol{\theta}_k^{(i)}(\text{blue circle}), \Delta \boldsymbol{x}_k^{(i)}(\text{red circle})\}$ and the cost value at $\{\boldsymbol{\theta}_k^{(i)}, \Delta \boldsymbol{x}_k^{(i)}\}$ is indicated by $H(\Delta \boldsymbol{x}_k^{(i)}, \boldsymbol{\theta}_k^{(i)})$ (the grey circle on the cost curve (the dark blue curve)). Our proofing strategy is as follows. First, we fix $\boldsymbol{\theta}_k^{(.)} \in \Omega_\theta$ and construct a neighborhood

$\mathcal{M}_k = \{\Omega_x, \boldsymbol{\theta}_k^{(\cdot)}\}$. Within this neighborhood we prove in Lemmas 2 and 4 that if we search for $\Delta\boldsymbol{x}_k^{(i)}$ through gradient ascent, we can converge to a local maximizer, and $H$ is maximizing with respect to $\Delta\boldsymbol{x}_k^{(i)}$. Second, we let $\Delta\boldsymbol{x}_k^{(\cdot)} \in \Omega_x$ be fixed, and we search for $\boldsymbol{\theta}_k^{(i)}$ through gradient descent. Under this condition, we demonstrate two ideas in Lemmas 3 and 5: (1) we show that $H$ is minimizing in the neighborhood $\mathcal{N}_k : \mathcal{N}_k = \{\Omega_\theta, \Delta\boldsymbol{x}_k^{(\cdot)}\}$; and (2) we converge to the local minimizer in the neighborhood $\mathcal{N}_k$. Third, in the union of the two neighborhoods $\mathcal{M}_k \cup \mathcal{N}_k$, (proven to be nonempty according to Lemma 6), we show that there exists at least one local equilibrium point (Theorem 1); that is, there is at least one balance point.

**Theorem 1** (Existence of an Equilibrium Point). *For any $k \in [0, K]$, let $\boldsymbol{\theta}_k^{(*)} \in \Omega_\theta$, be the minimizer of $H$ according to Lemma 5 and define $\mathcal{M}_k^{(*)} = \{\Omega_x, \boldsymbol{\theta}_k^{(*)}\}$. Similarly, let $\Delta\boldsymbol{x}_k^{(*)} \in \Omega_x$, be the maximizer of $H$ according to Lemma 4 and define $\mathcal{N}_k^{(*)} = \{\Delta\boldsymbol{x}_k^{(*)}, \Omega_\theta\}$. Further, let $\mathcal{M}_k^{(*)} \cup \mathcal{N}_k^{(*)}$ be nonempty according to Lemma. 6, then $(\Delta\boldsymbol{x}_k^{(*)}, \boldsymbol{\theta}_k^{(*)}) \in \mathcal{M}_k^{(*)} \cup \mathcal{N}_k^{(*)}$ is a local equilibrium point.*

We next show that this equilibrium point is stable (Theorem 2) under a sequential play. Specifically, we show that when player 1 plays first and player 2 plays second, we asymptotically reach a saddle point pair $(\Delta\boldsymbol{x}_k^{(*)}, \boldsymbol{\theta}_k^{(*)})$ for $H$. At this saddle point, both players have no incentive to move, and the game converges.

**Theorem 2** (Stability of the Equilibrium Point). *For any $k \in [0, K]$, $\Delta\boldsymbol{x}_k^{(i)} \in \Omega_x$ and $\boldsymbol{\theta}_k^{(i)} \in \Omega_\theta$ be the initial values for $\Delta\boldsymbol{x}_k^{(i)}$ and $\boldsymbol{\theta}_k^{(i)}$ respectively. Define $\mathcal{M}_k = \{\Omega_x, \Omega_\theta\}$ with $H(\Delta\boldsymbol{x}_k^{(i)}, \boldsymbol{\theta}_k^{(i)})$ given by Proposition 2. Let $\Delta\boldsymbol{x}_k^{(i+1)} - \Delta\boldsymbol{x}_k^{(i)} = \alpha_k^{(i)} \times (\nabla_{\Delta\boldsymbol{x}_k^{(i)}} H(\Delta\boldsymbol{x}_k^{(i)}, \boldsymbol{\theta}_k^{(\cdot)}))/\|\nabla_{\Delta\boldsymbol{x}_k^{(i)}} H(\Delta\boldsymbol{x}_k^{(i)}, \boldsymbol{\theta}_k^{(\cdot)})\|^2)$ and $\boldsymbol{\theta}_k^{(i+1)} - \boldsymbol{\theta}_k^{(i)} = -\alpha_k^{(i)} \times \nabla_{\boldsymbol{\theta}_k^{(i)}} H(\Delta\boldsymbol{x}_k^{(\cdot)}, \boldsymbol{\theta}_k^{(i)})$. Let the existence of an equilibrium point be given by Theorem 1, then, as a consequence of Lemmas 2 and 3, $(\Delta\boldsymbol{x}_k^{(*)}, \boldsymbol{\theta}_k^{(*)}) \in \mathcal{M}_k$ is a stable equilibrium point for $H$ given.*

In this game, the interplay between these two opposing players (representative of generalization and forgetting, respectively) introduces the dynamics required to play the game. Furthermore, the results presented in this section are local to the task. In other words, we prove that we can achieve a balance between generalization and forgetting for each task $k$ (neighborhoods are task dependent, and we achieve a local solution given a task $k$). Furthermore, our game is sequential; that is, there is a leader (player 1) and a follower (player 2). The leader ($\Delta\boldsymbol{x}_k^{(i)}$) plays first, and the follower ($\boldsymbol{\theta}_k^{(i)}$) plays second with complete knowledge of the leader's play. The game is directed by $\Delta\boldsymbol{x}_k^{(i)}$, and any changes in the task (reflected in $\Delta\boldsymbol{x}_k^{(i)}$) will shift the input and thus the equilibrium point. Consequently, the equilibrium point varies with respect to a task, and one will need to attain a new equilibrium point for each shift in a task. Without complete knowledge of the tasks (not available in a CL scenario), only a local result is possible. This highlights one of the key limitations of this work. Ideally, we would like a balance between forgetting and generalization that is independent of tasks. However, this would require learning a trajectory of the equilibrium point (How does the equilibrium point change with the change in the tasks?) and is beyond the scope of this paper. One work that attempts to do this is [39], where the authors learn a parameter per task. For a large number of tasks, however, such an approach is computationally prohibitive.

These results are valid only under certain assumptions: (1) the Frobenius norm of the gradient is bounded, always positive; (2) the cost function is bounded and differentiable; and (3) the learning rate goes to zero as $i$ tends to infinity. The first assumption is reasonable in practice, and gradient clipping or perturbation strategies can be used to ensure it. The boundedness of the cost (second assumption) can be ensured by prioritizing the contributions of the task (Lemma 1 and Corollary 1). The third assumption assumes a decaying learning rate. Learning rate decay is a common strategy and is employed widely. Therefore, all assumptions are practical and reasonable.

## 3.2 Balanced Continual Learning

Equipped with the theory, we develop a new CL method to achieve a balance between forgetting and generalization. By Proposition 2, the cost function can be upper bounded as $H(\Delta\boldsymbol{x}_k^{(i)}, \boldsymbol{\theta}_k^{(i)}) \leq$

$\beta_k J_k(\boldsymbol{\theta}_k^{(i)}) + (J_k(\boldsymbol{\theta}_k^{(i+\zeta)}) - J_k(\boldsymbol{\theta}_k^{(i)})) + (J_{k+\zeta}(\boldsymbol{\theta}_k^{(i)}) - J_k(\boldsymbol{\theta}_k^{(i)}))$, where $J_{k+\zeta}$ indicates $\zeta$ updates on player 1 and $\boldsymbol{\theta}_k^{(i+\zeta)}$ indicates $\zeta$ updates on player 2.

The strategies for the two players $\Delta\boldsymbol{x}_k, \boldsymbol{\theta}_k$ are chosen in Eq. (5) with $E$ being the expected value operator. We can approximate the required terms in our update rule (player strategies) using data samples (batches). Note that the approximation is performed largely through one-sided finite difference, which may introduce an error and is another potential drawback. The pseudo code of the BCL is shown in Algorithm 1. We define a new task array $\boldsymbol{\mathcal{D}}_N(k)$ and a task

$$
\underbrace{\frac{\alpha_k^{(i)} \nabla_{\Delta\boldsymbol{x}_k} E[H(\Delta\boldsymbol{x}_k^{(i)}, \boldsymbol{\theta}_k^{(i)})]}{\|\nabla_{\Delta\boldsymbol{x}_k} H(\Delta\boldsymbol{x}_k^{(i)}, \boldsymbol{\theta}_k^{(i)})\|^2}}_{\text{Player 1}},
$$

$$
\underbrace{-\alpha_k^{(i)} \times \nabla_{\boldsymbol{\theta}_k} E[H(\Delta\boldsymbol{x}_k^{(*)}, \boldsymbol{\theta}_k^{(i)}))]}_{\text{Player 2}},
$$

(5)

memory array $\boldsymbol{\mathcal{D}}_P(k) \subset \cup_{\tau=0}^{k-1} \boldsymbol{\mathcal{Q}}_\tau$ (samples from all previous tasks). For each batch $b_N \in \boldsymbol{\mathcal{D}}_N(k)$, we sample $b_P$ from $\boldsymbol{\mathcal{D}}_P(k)$, combine to create $b_{PN}(k) = b_P(k) \cup b_N(k)$, and perform a sequential play. Specifically, for each task the first player initializes $x_k^{PN} = b_{PN}(k)$ and performs $\zeta$ updates on $x_k^{PN}$ through gradient ascent. The second player, with complete knowledge of the first player's strategy, chooses the best play to reduce $H(\Delta\boldsymbol{x}_k^{(i)}, \boldsymbol{\theta}_k^{(i)})$. To estimate player 2's play, we must estimate different terms in $H(\Delta\boldsymbol{x}_k^{(i)}, \boldsymbol{\theta}_k^{(i)})$. This procedure involves three steps. First, we use the first player's play and approximate $(J_{k+\zeta}(\boldsymbol{\theta}_k^{(i)}) - J_k(\boldsymbol{\theta}_k^{(i)}))$. Second, to approximate $(J_k(\boldsymbol{\theta}_k^{(i+\zeta)}) - J_k(\boldsymbol{\theta}_k^{(i)}))$ : (a), we copy $\hat{\boldsymbol{\theta}}$ into $\hat{\boldsymbol{\theta}}_B$ (a temporary network) and perform $\zeta$ updates on $\hat{\boldsymbol{\theta}}_B$; and (b) we compute $J_k(\boldsymbol{\theta}_k^{(i+\zeta)})$ using $\hat{\boldsymbol{\theta}}_B(k+\zeta)$ and evaluate $(J_k(\boldsymbol{\theta}_k^{(i+\zeta)}) - J_k(\boldsymbol{\theta}_k^{(i)}))$. Third, equipped with these approximations, we compute $H(\Delta\boldsymbol{x}_k^{(i)}, \boldsymbol{\theta}_k^{(i)})$ and obtain the play for the second player. Both these players perform the steps repetitively for each piece of information (batch of data). Once all the data from the new task is exhausted, we move to the next task.

### 3.3 Related Work

Traditional solutions to the CL focus on either the forgetting issue [39, 49, 47, 7, 27, 50, 2, 31, 32, 11] or the generalization issue [45, 42, 18, 19, 8]. Common solutions to the forgetting problem involve dynamic architectures and flexible knowledge representation such as [39, 49, 47, 7], regularization approaches including [27, 50, 2] and memory/experience replay [31, 32, 11]. Similarly, quick generalization to a new task has been addressed through few-shot and one-shot learning approaches such as matching nets [45] and prototypical network [42]. More recently, the field of metalearning has approached the generalization problem by designing a metalearner that can perform quick generalization from very little data [18, 19, 8].

In the past few years, metalearners for quick generalization have been combined with methodologies specifically designed for reduced forgetting [22, 3]. For instance, the approaches in [22, 3] adapt the model-agnostic metalearning (MAML) framework in [18] with robust representation to minimize forgetting and provide impressive results on CL. However, both these approaches require a pretraining phase for learning representation. Simultaneously, Gupta et al. [20] introduced LA-MAML—a metalearning approach where the impact of learning rates on the CL problem is reduced through the use of per-parameter learning rates.

---

**Algorithm 1:** BCL

Initialize $\boldsymbol{\theta}, D_P, D_N$
**while** $k = 1, 2, 3, ...K$ **do**
    j = 0
    **while** $j < \rho$ **do**
        Get $\boldsymbol{b}_N \in D_k^N$
        Get $\boldsymbol{b}_P \in D_k^P$
        Get $\boldsymbol{b}_{PN} = \boldsymbol{b}_P \cup \boldsymbol{b}_N$
        Copy $\boldsymbol{b}_{PN}$ into $\boldsymbol{x}_k^{PN}$
        i = 0 **while** $i + 1 <= \zeta$ **do**
            Update $\boldsymbol{x}_k^{PN}$ with $J_k(\boldsymbol{\theta}_k)$ using gradient ascent
            i = i+1
        Calculate $J_{k+\zeta}(\boldsymbol{\theta}_k^{(i)}) - J_k(\boldsymbol{\theta}_k^{(i)})$
        Copy $\boldsymbol{\theta}_k^{(i)}$ into $\boldsymbol{\theta}_k^B$
        i = 0 **while** $i + 1 <= \zeta$ **do**
            Update $\boldsymbol{\theta}_k^B$ with $J_k(\boldsymbol{\theta}_k^B)$
            i = i+1
        Calculate $(J_k(\boldsymbol{\theta}_k^B) - J_k(\boldsymbol{\theta}_k^{(i)}))$
        Calculate $H(\Delta\boldsymbol{x}_k^{(i)}, \boldsymbol{\theta}_k^{(i)})$
        Update $\boldsymbol{\theta}_k^{(i)}$ using gradient descent
    j= j+1
Update $D_P$ with $D_N$

---

LA-MAML [20] also introduced episodic memory to address the forgetting issue. Other approaches also have attempted to model both generalization and forgetting. In [16], the gradients from new tasks are projected onto a subspace that is orthogonal to the older tasks, and forgetting is minimized. Similarly, Joseph and Balasubramanian [23] utilized a Bayesian framework to consolidate learning across previous and current tasks, and Yin et al. [48] provided a framework for approx-

imating loss function to summarize the forgetting in the CL setting. Furthermore, Abolfathi et al. [1] focused on sampling episodes in the reinforcement learning setting, and Elrahimi et al. [15] introduced a generative adversarial network-type structure to progressively learn shared features assisting reduced forgetting and improved generalization. Despite significant progress, however, these methods [22, 3, 20, 16, 23, 48, 1, 15] are still inherently tilted toward maximizing generalization or minimizing forgetting because they naively minimize the loss function. Therefore, the contribution of different terms in the loss function becomes important. For instance, if the generalization cost is given more weight, a method would generalize better. Similarly, if forgetting cost is given more weight, a method would forget less. Therefore, the resolution of the trade-off inherently depends on an hyperparameter.

The first work to formalize the trade-off in CL was MER, where the trade-off was formalized as a gradient alignment problem. Similar to MER, Doan et al. ([13]) studied forgetting as an alignment problem. In MER, the angle between the gradients was approximated by using Reptile [36], which promotes gradient alignment by reducing weight changes. On the other hand, [13] formalized the alignment as an eigenvalue problem and introduced a PCA-driven method to ensure alignment. Our approach models this balance as a saddle point problem achieved through stochastic gradient such that the saddle point (balance point or equilibrium point) resolves the trade-off.

Our approach is the first in the CL literature to prove the existence of the saddle point (the balance point) between generalization and forgetting given a task in a CL problem. Furthermore, we are the first to theoretically demonstrate that the saddle point can be achieved reasonably under a gradient ascent-descent game. The work closest to ours is [15], where an adversarial framework is described to minimize forgetting in CL by generating task-invariant representation. However, [15] is not model agnostic (the architecture of the network is important) and requires a considerable amount of data at the start of the learning procedure. Because of these issues, [15] is not suitable for learning in the sequential scenario.

Several attempts have been reported in the literature to theoretically analyze different aspects of CL. For instance, Benzing [6] attempted to unify regularization methods such as [27, 50, 2, 31]. On the other hand, Benamin et al. [5] provided generalization (in the context of training to test data generalization) guarantees with orthogonal gradient descent, and Yin et al. [48] provided generalization (in the context of training to test data generalization) analysis while regularizing with a second-order approximation of the loss functions. Although these works provide important results, they focus on quantifying catastrophic forgetting or generalization but do attempt to model the trade-off in any manner.

# 4 Experiments

We use the CL benchmark [21] for our experiments and retain the experimental settings (hyperparameters) from [21, 44]. For comparison, we use the split-MNIST, permuted-MNIST, and split-CiFAR100 data sets while considering three scenarios: incremental domain learning (IDL), incremental task learning (ITL), and incremental class learning (ICL). The splitting and permutation strategies when applied to the MNIST or CiFAR100 data set can generate task sequences for all three scenarios (illustrated in Figure 1 and Appendix: Figure 2 of [21]). For comparing our approach, we use three baseline strategies—standard neural network with Adam [26], Adagrad [14], and SG—and use $L_2$-regularization and naive rehearsal (which is similar to experience replay). For CL approaches, we use EWC [27], online EWC [40], SI [50], LwF [30], DGR [41], RtF [44], MAS [2], MER [38], and GEM [33]. We utilize data preprocessing as provided by [21]. Additional details on experiments can be found in Appendix B and [21, 44]. All experiments are conducted in Python 3.4 using the pytorch 1.7.1 library with the NVIDIA-A100 GPU for our simulations.

**Comparison with the state of the art:** The results for our method are summarized in Table 1 and 2. The efficiency for any method is calculated by observing the average accuracy (retained accuracy (RA) [38]) at the end of each repetition and then evaluating the mean and standard deviation of RA across different repetitions. For each method, we report the mean and standard deviation of RA over five repetitions of each experiment. In each column, we indicate the best-performing method in bold. For the split-MNIST data set, we obtain $99.52 \pm 0.07$ for ITL, $98.71 \pm 0.06$ for IDL, and $97.32 \pm 0.17$ for ICL. Similarly, with the permuted-MNIST data set, we obtain $97.41 \pm 0.01$ for ITL, $97.51 \pm 0.05$ for IDL, and $97.61 \pm 0.01$ for ICL. Furthermore, with the split-CiFAR100 data

Table 1: Performance of our approach compared with other methods in the literature. We record the mean and standard deviation of the retained accuracy for the different methods. The best scores are in bold.

| Method | split-MNIST | | | permuted-MNIST | | |
|---|---|---|---|---|---|---|
| | Incremental task learning [ITL] | Incremental domain learning [IDL] | Incremental class learning [ICL] | Incremental task learning [ITL] | Incremental domain learning [IDL] | Incremental class learning [ICL] |
| Adam | $95.52 \pm 2.14$ | $54.75 \pm 2.06$ | $19.72 \pm 0.03$ | $93.42 \pm 0.56$ | $77.87 \pm 1.27$ | $14.02 \pm 1.25$ |
| SGD | $97.65 \pm 0.28$ | $62.80 \pm 0.34$ | $19.36 \pm 0.02$ | $90.95 \pm 0.20$ | $78.17 \pm 1.16$ | $12.82 \pm 0.95$ |
| Adagrad | $98.37 \pm 0.29$ | $57.59 \pm 2.54$ | $19.59 \pm 0.17$ | $92.45 \pm 0.16$ | $91.59 \pm 0.46$ | $29.09 \pm 1.48$ |
| $L_2$ | $97.62 \pm 0.69$ | $66.84 \pm 3.91.$ | $22.92 \pm 1.90$ | $94.87 \pm 0.38$ | $92.81 \pm 0.32$ | $13.92 \pm 1.79$ |
| Naive rehearsal | $99.32 \pm 0.10$ | $94.85 \pm 0.80$ | $90.88 \pm 0.70$ | $96.23 \pm 0.04$ | $95.84 \pm 0.06$ | $96.25 \pm 0.10$ |
| Naive rehearsal-C | $99.41 \pm 0.04$ | $97.13 \pm 0.37$ | $94.92 \pm 0.63$ | $97.13 \pm 0.03$ | $96.75 \pm 0.03$ | $97.24 \pm 0.05$ |
| EWC | $96.59 \pm 0.99$ | $57.31 \pm 1.07$ | $19.70 \pm 0.14$ | $95.38 \pm 0.33$ | $89.54 \pm 0.52$ | $26.32 \pm 4.32$ |
| Online EWC | $99.01 \pm 0.12$ | $58.25 \pm 1.23$ | $19.68 \pm 0.05$ | $95.15 \pm 0.49$ | $93.47 \pm 0.01$ | $42.58 \pm 6.50$ |
| SI | $99.10 \pm 0.16$ | $64.63 \pm 1.67$ | $19.67 \pm 0.25$ | $94.35 \pm 0.51$ | $91.12 \pm 0.93$ | $58.52 \pm 4.20$ |
| MAS | $98.88 \pm 0.14$ | $61.98 \pm 7.17$ | $19.70 \pm 0.34$ | $94.74 \pm 0.52$ | $93.22 \pm 0.80$ | $50.81 \pm 2.92$ |
| GEM | $98.32 \pm 0.08$ | $97.37 \pm 0.22$ | $93.04 \pm 0.05$ | $95.44 \pm 0.96$ | $96.86 \pm 0.02$ | $96.72 \pm 0.03$ |
| DGR | $99.47 \pm 0.03$ | $95.74 \pm 0.23$ | $91.24 \pm 0.33$ | $92.52 \pm 0.08$ | $95.09 \pm 0.04$ | $92.19 \pm 0.09$ |
| RtF | $\mathbf{99.66 \pm 0.03}$ | $97.31 \pm 0.11$ | $92.56 \pm 0.21$ | $97.31 \pm 0.01$ | $97.06 \pm 0.02$ | $96.23 \pm 0.04$ |
| MER | $97.12 \pm 0.10$ | $92.16 \pm 0.35$ | $93.20 \pm 0.12$ | $97.15 \pm 0.08$ | $96.11 \pm 0.31$ | $91.71 \pm 0.03$ |
| BCL (With Game) | $99.52 \pm 0.07$ | $\mathbf{98.71 \pm 0.06}$ | $\mathbf{97.32 \pm 0.17}$ | $\mathbf{97.41 \pm 0.01}$ | $\mathbf{97.51 \pm 0.05}$ | $\mathbf{97.61 \pm 0.01}$ |
| BCL (Without Game) | $97.73 \pm 0.03$ | $96.43 \pm 0.29$ | $91.88 \pm 0.55$ | $96.16 \pm 0.03$ | $96.08 \pm 0.06$ | $95.96 \pm 0.06$ |

set, we obtain $81.82 \pm 0.17$ for ITL, $62.11 \pm 0.00$ for IDL, and $69.27 \pm 0.03$ for ICL. BCL is the best-performing methodology for all cases (across both data sets) except RtF for ITL ($0.14\%$ drop) with the split-MNIST data set.

Generally, ITL is the easiest learning scenario [21], and all methods therefore perform well on ITL (the performance is close). For the ITL scenario with the split-MNIST data set, BCL is better than most methods; but several methods, such as naive rehearsal, naive rehearsal-C, RtF, and DGR, attain close RA values (less than $1\%$ from BCL). Note that both DGR and RtF involve a generative model pretrained with data from all the tasks. In a sequential learning scenario, one cannot efficiently train generative models because data corresponding to all the tasks is not

Table 2: Performance of BCL for the split-CiFAR100 data set. We record the retained accuracy for the different methods. We obtained RA scores for all methods except BCL from [21].

| Method | split-CiFAR100 | | |
|---|---|---|---|
| | Incremental task learning | Incremental domain learning | Incremental class learning |
| Adam | $30.53 \pm 0.58$ | $19.65 \pm 0.14$ | $17.20 \pm 0.06$ |
| SGD | $43.77 \pm 1.15$ | $19.17 \pm 0.12$ | $17.18 \pm 0.12$ |
| Adagrad | $36.27 \pm 0.43$ | $19.06 \pm 0.14$ | $15.83 \pm 0.20$ |
| $L_2$ | $51.73 \pm 1.30$ | $19.96 \pm 0.15$ | $17.12 \pm 0.04$ |
| Naive rehearsal | $70.20 \pm 0.17$ | $35.94 \pm 0.39$ | $34.33 \pm 0.19$ |
| Naive rehearsal-C | $78.41 \pm 0.37$ | $51.81 \pm 0.18$ | $51.28 \pm 0.17$ |
| EWC | $61.11 \pm 1.43$ | $19.76 \pm 0.12$ | $19.70 \pm 0.14$ |
| Online EWC | $63.22 \pm 0.97$ | $20.03 \pm 0.10$ | $17.16 \pm 0.09$ |
| SI | $64.81 \pm 1.00$ | $20.26 \pm 0.09$ | $17.26 \pm 0.11$ |
| MAS | $64.77 \pm 0.78$ | $19.99 \pm 0.16$ | $17.07 \pm 0.12$ |
| BCL(With Game) | $\mathbf{81.82 \pm 0.17}$ | $\mathbf{62.11 \pm 0.00}$ | $\mathbf{69.27 \pm 0.03}$ |
| BCL(Without Game) | $69.17 \pm 0.12$ | $51.82 \pm 0.19$ | $52.82 \pm 0.01$ |

available beforehand. Although RtF provides improved performance for split-MNIST (ITL), the improvement is less than $1\%$ (not significant). In fact, RtF performance is poorer for BCL in ICL by $4.76\%$ (a significant drop in performance) and in IDL by $1.4\%$.

Two additional observations can be made about the split-MNIST data set. First, Adagrad, SGD, and L2 achieve better performance than does Adam. Therefore, in our analysis Adagrad appears more appropriate although Adam is popularly used for this task. Second, naive rehearsal (both naive rehearsal and naive rehearsal-C approaches) achieves performance equivalent to the state-of-the-art methods with similar memory overhead. Furthermore, naive rehearsal performs much better than online EWC and SI, especially in the ICL scenario. These limitations indicate that regularization-driven approaches are not much better than baseline models and in fact perform poorer than methods involving memory (naive rehearsals, MER, BCL, etc.). In [28], it was shown theoretically that memory-based approaches typically do better than regularization-driven approaches, as is empirically observed in this paper, too. Another interesting observation is that EWC and online EWC require significant hyperparameter tuning, which would be difficult to do in real-world scenarios. Other regularization-based methods, such as SI and MAS, also suffer from the same issue.

The observations from the split-MNIST carry forward to the permuted-MNIST data set. Moreover, RA values for the permutation MNIST data set are better for the split-MNIST data set across the board, indicating that the permutation MNIST data set presents an easier learning problem. Similar to the observations made with the split-MNIST data set, BCL is better than all methods for the permuted-MNIST dataset, with naive rehearsal and RtF providing RA values that are close (less than $1\%$). The only methodology in the literature that attempts to model the balance between forgetting and generalization is MER, an extension of GEM. From our results, we observe that BCL is better than

MER in all cases (Split-MNIST–2.4% improvement for ITL, 6.55% improvement for IDL, 4.12% improvement for ICL and Permuted-MNIST–0.26% improvement for ITL, 1.4% improvement for IDL and 5.9% improvement for ICL).

The substantial improvements obtained by BCL are more evident in Table 2 where the results on the split-CiFAR100 data set are summarized. BCL is clearly the best-performing method. The next best-performing method is naive rehearsal-C, where BCL improves performance by 3.41% for ITL, 10.3% for IDL, and 17.99% for ICL. Other observations about the regularization methods and the rest of the baseline methods carry forward from Table 1. However, one key difference is that while AdaGrad is observed to be better than Adam for the MNIST data set, Adam is comparable to SGD and Adagrad for split-CiFAR100. In summary, BCL is comparable to or better than the state of the art in the literature for both the MNIST and CiFAR100 data sets.

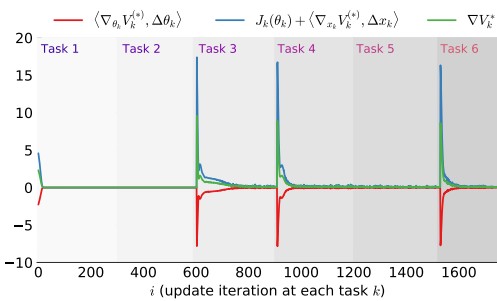

(a)

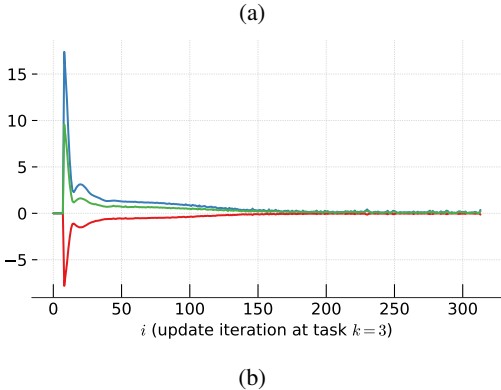

(b)

Figure 3: (Top) Progression of different terms in Eq. (2) with respect to update iterations, where the index on the x axis is calculated as $k \times 300$. The tasks boundaries (at what $i$ the tasks are introduced) are illustrated through shades of grey. (down) Illustration of the cost at $k = 3$.

**Do we need a game to achieve this performance?** To provide additional insights into the benefits of the game, we compare RA values with and without the game. In our setup, $\Delta \boldsymbol{x}_k^{(i)}$ aims at increasing the cost, and $\boldsymbol{\theta}_k^{(i)}$ aims at reducing the cost. If we hold the play for $\Delta \boldsymbol{x}_k^{(i)}$, then the dynamics required to play the game do not exist, thus providing a method that can perform CL without the game. Therefore, we induce the absence of a game by fixing $\Delta \boldsymbol{x}_k^{(i)}$ and perform each of the nine experiments for five repetitions (ICL, IDL, and ITL for split-MNIST, permuted-MNIST, and split-CiFAR100). We summarize these results in the last two rows of Tables 1 and 2. Consequently, we make two observations. First: even without the game, we achieve RA values comparable to the state of the art. This is true for both the MNIST and CiFAR100 data sets. Second, with the introduction of the game, we observe improved RA values across the board (at least by 1% with MNIST). The difference is clearer with the CiFAR100 data set where we observe a substantial improvement in RA values (at least 10%).

**Does the theory appropriately model the continual learning problem?** In Fig. 3a we plot the progression of $J_k(\boldsymbol{\theta}_k) + \langle \nabla_{\boldsymbol{x}_k} V_k^{(*)}, \Delta \boldsymbol{x}_k \rangle$ (blue curve), $\langle \nabla_{\boldsymbol{\theta}_k} V_k^{(*)}, \Delta \boldsymbol{\theta}_k \rangle$ (red curve), and the sum, namely, $\Delta V_k^{(*)}$ (green curve). A total of six tasks, sampled from the permutation MNIST data set within the ICL setting, are illustrated in Fig. 3a. These tasks are introduced every 300 update steps. From Fig. 3a we make two important observations. First, as soon as tasks 1, 3, 4, and 6 are introduced, all three curves (red, blue, and green) indicate a bump. Second, when tasks 2 and 5 are introduced, there is no change. On a closer look at task 3 in Fig. 3b, we observe that when the task 3 is introduced, the blue curve exhibits a large positive bump (the introduction of the new tasks increases the first and the third terms in Eq. (2)). The increase implies that task 3 forced the model to generalize and increased forgetting on tasks 1 and 2 (observed by the increase in the green curve). To compensate for this increase, we require the model $\boldsymbol{\theta}_k$ to behave adversarially and introduce a large enough negative value in $\langle \nabla_{\boldsymbol{\theta}_k} V_k^{(*)}, \Delta \boldsymbol{\theta}_k \rangle$ (red curve) to cancel out the increase in the blue curve. In Fig. 3b the red curve demonstrates a large negative value (expected behavior) and eventually (as $i$ increases) forces the blue curve (by consequence, green: the sum of red and blue) to move toward zero (the model compensates for the increase in forgetting). As observed, the blue and the red curves behave opposite to each other and introduce a push-pull behavior that stops only when the two cancel each other and the sum (green) is zero. Once the sum has reached zero, there is no incentive for the

red and green to be nonzero, and therefore they remain at zero; thus, all three curves (green, red, and blue) remain at zero once converged until a task 4 is introduced (when there is another bump, as seen in Fig. 3a). However, this increase in the blue curve is not observed when tasks 2 and 5 are introduced. When new tasks are similar to the older tasks, it is expected that $\Delta V_k^{(*)} = 0$, as is observed in Fig. 3a.

All of these observations are fully explained by Eq. 2, which illustrates that the solution to the CL problem is obtained optimally only when $\Delta V_k^{(*)} = 0$ (observed in Figs. 3a and 3b). The term $\Delta V_k^{(*)}$ is quantified by $J_k(\boldsymbol{\theta}_k)$, $\langle \nabla_{\boldsymbol{\theta}_k} V_k^{(*)}, \Delta\boldsymbol{\theta}_k \rangle$ and $\langle \nabla_{\boldsymbol{x}_k} V_k^{(*)}, \Delta\boldsymbol{x}_k \rangle$. Our theory suggests that there exists an inherent trade-off between different terms in Eq. 2. Therefore every time a new task is observed, it is expected that $J_k(\boldsymbol{\theta}_k) + \langle \nabla_{\boldsymbol{x}_k} V_k^{(*)}, \Delta\boldsymbol{x}_k \rangle$ increases (increase in blue curve when tasks 1, 3, 4, and 6 are introduced) and $\langle \nabla_{\boldsymbol{\theta}_k} V_k^{(*)}, \Delta\boldsymbol{\theta}_k \rangle$ compensates to cancel this increase (red curve exhibits a negative jump). In Theorems 1 and 2 we demonstrate the existence of this balance point (for each task, as $i$ increases, $\Delta V_k^{(*)}$ tends to zero, as observed in Fig. 3b) and $\Delta V_k^{(*)}$ remains zero (the balance point is stable, proved in Theorem 2) until a new task increases forgetting. Furthermore, our theory claims the existence of a solution with respect to each task. This is also observed in Fig. 3a as, for each task, there is an increase in cost, and BCL quickly facilitates convergence. *These observations indicate that our dynamical system in Eq. 2 accurately describes the dynamics of the continual learning problem.* Furthermore, the assumptions under which the theory is developed are practical and are satisfied by performing continual learning on the permuted MNIST problem.

## 5    Conclusion

We developed a dynamic programming-based framework to enable the methodical study of key challenges in CL. We show that an inherent trade-off between generalization and forgetting exists and must be modeled for optimal performance. To this end, we introduce a two-player sequential game that models the trade-off. We show in theory and simulation that there is an equilibrium point that resolves this trade-off (Theorem 1) and that this saddle point can be attained (Theorem 2). However, we observe that any change in the task modifies the equilibrium point. Therefore, a global equilibrium point between generalization and forgetting is not possible, and our results are valid only in a neighborhood (defined given a task). To attain this equilibrium point, we develop BCL and demonstrate state-of-the-art performance on a CL benchmark [21]. In the future, we will extend our framework for nonEuclidean tasks.

## 6    Broader Impact

*Positive Impacts:* CL has a wide range of applicability. It helps avoids retraining, and it improves the learning efficiency of learning methods. Therefore, in science applications where the data is generated sequentially but the data distribution varies with time, our theoretically grounded method provides the potential for improved performance. *Negative Impacts:* Our theoretical framework does not have direct adverse impacts. However, the potential advantages of our approach can improve the efficiency of adverse ML systems such as fake news, surveillance, and cybersecurity attacks.

## Acknowledgments and Disclosure of Funding

This work was supported by the U.S. Department of Energy, Office of Science, Advanced Scientific Computing Research, under Contract DE-AC02-06CH11357 and by a DOE Early Career Research Program award. We are grateful for the computing resources from the Joint Laboratory for System Evaluation and Leadership Computing Facility at Argonne. We also are grateful to Dr. Vignesh Narayanan, assistant professor, University of South Carolina, and Dr. Marieme Ngom, Dr. Sami Khairy –postdoctoral appointees, Argonne National Laboratory, for their insights.

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
