# Formalizing the Generalization-Forgetting Trade-Off in Continual Learning

**R. Krishnan**[1] **and Prasanna Balaprakash**[1,2]
[1]Mathematics and Computer Science Division
[2]Leadership Computing Facility
Argonne National Laboratory
*kraghavan,pbalapra@anl.gov*

## Abstract

We formulate the continual learning problem via dynamic programming and model the trade-off between catastrophic forgetting and generalization as a two-player sequential game. In this approach, player 1 maximizes the cost due to lack of generalization whereas player 2 minimizes the cost due to increased catastrophic forgetting. We show theoretically and experimentally that a balance point between the two players exists for each task and that this point is stable (once the balance is achieved, the two players stay at the balance point). Next, we introduce balanced continual learning (BCL), which is designed to attain balance between generalization and forgetting, and we empirically demonstrate that BCL is comparable to or better than the state of the art.

## Supplementary Information

We use $\mathbb{R}$ to denote the set of real numbers and $\mathbb{N}$ to denote the set of natural numbers. We use $\|.\|$ to denote the Euclidean norm for vectors and the Frobenius norm for matrices, while using bold symbols to illustrate matrices and vectors. We define an interval $[0, K), K \in \mathbb{N}$ and let $p(\mathcal{Q})$ be the distribution over all the tasks observed in this interval. For any $k \in [0, K)$, we define a parametric model $g(.)$ with $\boldsymbol{y}_k = g(\boldsymbol{x}_k; \boldsymbol{\theta}_k)$, where $\boldsymbol{\theta}_k$ is a vector comprising all parameters of the model with $\boldsymbol{x}_k \in \mathcal{X}_k$. Let $n$ be the number of samples and $m$ be the number of dimensions. Suppose a task at $k|k \in [0, K)$ is observed and denoted as $\mathcal{Q}_k : \mathcal{Q}_k \sim p(\mathcal{Q})$, where $\mathcal{Q}_k = \{\mathcal{X}_k, \ell_k\}$ is a tuple with $\mathcal{X}_k \in \mathbb{R}^{nm}$ being the input data and $\ell_k$ quantifies the loss incurred by $\mathcal{X}_k$ using the model $g$ for the task at $k$. We denote a sequence of $\boldsymbol{\theta}_k$ as $\boldsymbol{u}_{k:K} = \{\boldsymbol{\theta}_\tau \in \Omega_\theta, k \leq \tau \leq K\}$, with $\Omega_\theta$ being the compact (feasible) set for the parameters. We denote the optimal value with a superscript $(*)$; for instance, we use $\boldsymbol{\theta}_k^{(*)}$ to denote the optimal value of $\boldsymbol{\theta}_k$ at task $k$. In this paper we use balance point, equilibrium point, and saddle point to refer to the point of balance between generalization and forgetting. We interchange between these terms whenever convenient for the discussion. We will use $\nabla_{(j)} i$ to denote the gradient of $i$ with respect to $j$ and $\Delta i$ to denote the first difference in discrete time.

## 1  Additional Results

We define the cost (combination of catastrophic cost and generalization cost) at any instant $k$ as $J_k(\boldsymbol{\theta}_k) = \gamma_k \ell_k + \sum_{\tau=0}^{k-1} \gamma_\tau \ell_\tau$, where $\ell_\tau$ is computed on task $\mathcal{Q}_\tau$ with $\gamma_\tau$ describing the contribution of $\mathcal{Q}_\tau$ to this sum. We will show that for any fixed $k$, the catastrophic forgetting cost $J_k(\boldsymbol{\theta}_k)$ is divergent in the limit $k \to \infty$ if equal contribution from each task is expected.

35th Conference on Neural Information Processing Systems (NeurIPS 2021).

**Lemma 1.** *For any $k \in \mathbb{N}$, define $J_k(\boldsymbol{\theta}_k) = \sum_{\tau=0}^{k} \gamma_\tau \ell_\tau$. For all $\tau$, assume $\ell_\tau$ to be continuous with $L \geq \ell_\tau \geq \epsilon, \forall \tau, \epsilon > 0$ and let $\gamma_\tau = 1$. Then $J_k(\boldsymbol{\theta}_k)$ is divergent as $k \to \infty$.*

*Proof of Lemma 1.* With $J_k(\boldsymbol{\theta}_k) = \sum_{\tau=0}^{k} \gamma_\tau \ell_\tau$ we write $\lim_{k\to\infty} \sum_{\tau=0}^{k} \gamma_\tau \ell_\tau \geq \lim_{k\to\infty} \sum_{\tau=0}^{k} \gamma_\tau \epsilon$, where $\gamma_\tau = 1$ which provides $\lim_{k\to\infty} \sum_{\tau=0}^{k} \epsilon = \infty$. Therefore, $J_k(\boldsymbol{\theta}_k)$ is divergent. $\square$

When $\ell_\tau \geq \epsilon$ with $\epsilon > 0$ implies that each task incurs a nonzero cost. Furthermore, $\gamma_\tau = 1$, it implies that each task provides equal contribution to the catastrophic forgetting cost and contributed nonzero value to $J_k(\boldsymbol{\theta}_k)$. The aforementioned lemma demonstrates that equivalent performance (no forgetting on all tasks ) cannot be guaranteed for an infinite number of tasks when each task provides a nonzero cost to the sum (you have to learn for all the tasks ). However, if the task contributions are prioritized based on knowledge about the task distribution, the sum can be ensured to be convergent as shown in the next corollary.

**Corollary 1.** *For any $k \in \mathbb{N}$, define $J_k(\boldsymbol{\theta}_k) = \sum_{\tau=0}^{k} \gamma_\tau \ell_\tau$ where $\ell_\tau$ is continuous and bounded such that $\epsilon \leq \ell_\tau \leq L, \forall \epsilon > 0$. Define $N = \frac{1}{k}$ and choose $\gamma_N$ such that $\gamma_N \to 0, N \to \infty$ and assume when there are infinite number of tasks, $\lim_{N\to\infty} \sum_N \gamma_N \leq M$. Under these assumptions, $J_k(\boldsymbol{\theta}_k)$ is convergent.*

*Proof of Corollary 1.* Since $\ell_\tau \leq L$, $J_k(\boldsymbol{\theta}_k) = \lim_{k\to\infty} \sum_{\tau=0}^{k} \gamma_\tau \ell_\tau \leq \lim_{k\to\infty} \sum_{\tau=0}^{k} \gamma_\tau L \leq L\lim_{k\to\infty} \sum_{\tau=0}^{k} \gamma_\tau$.

Since $\lim_{k\to\infty} \sum_{\tau=0}^{k} \gamma_\tau = \lim_{N\to\infty} \sum_N \gamma_N$ as $N = \frac{1}{k}$, therefore $\lim_{k\to\infty} \sum_{\tau=0}^{k} \gamma_\tau \leq M$ and $J_k(\boldsymbol{\theta}_k)$ is upper bounded by $LM$. As a result, $J_k(\boldsymbol{\theta}_k)$ is convergent since $J_k(\boldsymbol{\theta}_k)$ is a monotone. $\square$

To solve the problem at $k$, we seek $\boldsymbol{\theta}_k$ to minimize $J_k(\boldsymbol{\theta}_k)$. Similarly, to solve the problem in the complete interval $[0, K)$, we seek a $\boldsymbol{\theta}_k$ to minimize $J_k(\boldsymbol{\theta}_k)$ for each $k \in [0, K)$. In other words we seek to obtain $\boldsymbol{\theta}_k$ for each task such that the cost $J_k(\boldsymbol{\theta}_k)$ is minimized. The optimization problem for the overall CL problem (overarching goal of CL ) is then provided as the minimization of the cumulative cost $V_k(\boldsymbol{u}_{k:K}) = \sum_{\tau=k}^{K} \beta_\tau J_\tau(\boldsymbol{\theta}_\tau)$ such that $V_k^{(*)}$, is given as

$$V_k^{(*)} = min_{\boldsymbol{u}_{k:K}} V_k(\boldsymbol{u}_{k:K}), \tag{1}$$

with $0 \leq \beta_\tau \leq 1$ being the contribution of $J_\tau$ and $\boldsymbol{u}_{k:K}$ being a weight sequence of length $K - k$. We will now derive the difference equation for our cost formulation.

**Proposition 1.** *For any $k \in [0, K)$, define $V_k = \sum_{\tau=k}^{K} \beta_\tau J_\tau(\boldsymbol{\theta}_\tau)$ with $\boldsymbol{\theta}_\tau \in \Omega$. Define $\boldsymbol{u}_{k:K} = \{\boldsymbol{\theta}_\tau \in \Omega, k \leq \tau \leq K\}$,, with $\Omega$ being the compact (feasible ) set as a sequence of parameters with length $K - k$ and $V_k^{(*)} = min_{\boldsymbol{u}_{k:K}} \sum_{\tau=k}^{K} \beta_\tau J_\tau(\boldsymbol{\theta}_\tau)$. Then, the following is true*

$$\Delta V_k^{(*)} = -min_{\boldsymbol{\theta}_k \in \Omega} \big[\beta_k J_k(\boldsymbol{\theta}_k) + \big(\langle \nabla_{\boldsymbol{\theta}_k} V_k^{(*)}, \Delta\boldsymbol{\theta}_k \rangle + \langle \nabla_{\boldsymbol{x}_k} V_k^{(*)}, \Delta\boldsymbol{x}_k \rangle\big)\big], \tag{2}$$

*where $\Delta V_k^{(*)}$ represents the first difference due to the introduction of a task, $\Delta\boldsymbol{\theta}_k$ due to parameters and $\nabla_{\boldsymbol{x}_k}$ due to the task data with $\beta_k \in \mathbb{R} \cup [0, 1], \forall k$ and $J_k(\boldsymbol{\theta}_k) = \gamma_k \ell_k + \sum_{\tau=0}^{k-1} \gamma_\tau \ell_\tau$.*

*Proof.* Given $V_k^{(*)} = min_{\boldsymbol{u}_{k:K}} \sum_{\tau=k}^{K} \beta_\tau J_\tau(\boldsymbol{\theta}_\tau)$, we split the interval $[k, K)$ as $[k, k+1)$ and $[k+1, K)$ to write

$$V_k^{(*)} = min_{\boldsymbol{\theta}_\tau \in \Omega} \big[\beta_k J_k(\boldsymbol{\theta}_k)\big] + min_{\boldsymbol{u}_{k+1:K}} \big[ \sum_{\tau=k+1}^{K} \beta_\tau J_\tau(\boldsymbol{\theta}_\tau)\big].$$

$V_k = \sum_{\tau=k}^{K} \beta_\tau J_\tau(\boldsymbol{\theta}_\tau)$ provides $\sum_{\tau=k+1}^{K} \beta_\tau J_\tau(\boldsymbol{\theta}_\tau)$ is $V_{k+1}$ therefore $min_{\boldsymbol{u}_{k+1:K}} \big[\sum_{\tau=k+1}^{K} \beta_\tau J_\tau(\boldsymbol{\theta}_\tau)\big]$ is $V_{k+1}^{(*)}$. We then achieve

$$V_k^{(*)} = min_{\boldsymbol{\theta}_k \in \Omega} \big[\beta_k J_k(\boldsymbol{\theta}_k) + V_{k+1}^{(*)}\big].$$

Since the minimization is with respect to $k$ now, the terms in $k + 1$ can be pulled into of the bracket without any change to the minimization problem. We then approximate $V_{k+1}^{(*)}$ using the information provided at $k$. Since $V_{k+1}^{(*)}$ is a function of $\boldsymbol{y}_k$, which is then a function of $(k, \boldsymbol{x}_k, \boldsymbol{\theta}_k)$, and all changes in $\boldsymbol{y}_k$ can be summarized through $(k, \boldsymbol{x}_k, \boldsymbol{\theta}_k)$. Therefore, a Taylor series of $V_{k+1}^{(*)}$ around $(k, \boldsymbol{x}_k, \boldsymbol{\theta}_k)$ provides

$$
\begin{aligned}
V_{k+1}^{(*)} = V_k^{(*)} &+ \langle \nabla_{\boldsymbol{\theta}_k} V_k^{(*)}, \Delta \boldsymbol{\theta}_k \rangle \\
&+ \langle \nabla_{\boldsymbol{x}_k} V_k^{(*)}, \Delta \boldsymbol{x}_k \rangle + \langle \nabla_k (V_k^{(*)}), \Delta k \rangle + \cdots,
\end{aligned}
\tag{3}
$$

where $\cdots$ summarizes all the higher order terms. As $k \in \mathbb{N}$ and $\langle \nabla_k (V_k^{(*)}), \Delta k \rangle$ represents the first difference in $V_k^{(*)}$ hitherto denoted by $\Delta V_k^{(*)}$. We therefore achieve

$$
\begin{aligned}
V_{k+1}^{(*)} = V_k^{(*)} &+ \langle \nabla_{\boldsymbol{\theta}_k} V_k^{(*)}, \Delta \boldsymbol{\theta}_k \rangle \\
&+ \langle \nabla_{\boldsymbol{x}_k} V_k^{(*)}, \Delta \boldsymbol{x}_k \rangle + \Delta V_k^{(*)} + \quad,
\end{aligned}
\tag{4}
$$

Substitute into the original equation to get

$$
\begin{aligned}
V_k^{(*)} = min_{\boldsymbol{\theta}_k \in \Omega} \big[ \beta_k J_k(\boldsymbol{\theta}_k) \big] &+ \big( V_k^{(*)} + \langle \nabla_{\boldsymbol{\theta}_k} V_k^{(*)}, \Delta \boldsymbol{\theta}_k \rangle \\
&+ \langle \nabla_{\boldsymbol{x}_k} V_k^{(*)}, \Delta \boldsymbol{x}_k \rangle + \Delta V_k^{(*)} \big) + \cdots,
\end{aligned}
\tag{5}
$$

Cancel common terms and assume that the higher order terms $(\cdots)$ are negligible to obtain

$$
\Delta V_k^{(*)} = -min_{\boldsymbol{\theta}_k \in \Omega} \big[ \beta_k J_k(\boldsymbol{\theta}_k) + \langle \nabla_{\boldsymbol{\theta}_k} V_k^{(*)}, \Delta \boldsymbol{\theta}_k \rangle + \langle \nabla_{\boldsymbol{x}_k} V_k^{(*)}, \Delta \boldsymbol{x}_k \rangle \big].
\tag{6}
$$

which is a difference equation in $V_k^{(*)}$. $\qquad \square$

Note that $V_k^{(*)}$ is the minima for the overarching CL problem and $\Delta V_k^{(*)}$ represents the change in $V_k^{(*)}$ upon introduction of a task (we hitherto refer to this as perturbations). Zero perturbations $(\Delta V_k^{(*)} = 0)$ implies that the introduction of a new task does not impact our current solution; that is, the optimal solution on all previous tasks is optimal on the new task as well.

The solution of the CL problem can directly be obtained by solving Eq. (6) using all the available data. Thus, $min_{\boldsymbol{\theta}_k \in \Omega} \big[ H(\Delta \boldsymbol{x}_k, \boldsymbol{\theta}_k) \big]$ yields $\Delta V_k^{(*)} \approx 0$ for $\beta > 0$, with $H(\Delta \boldsymbol{x}_k, \boldsymbol{\theta}_k) = \beta_k J_k(\boldsymbol{\theta}_k) + \langle \nabla_{\boldsymbol{\theta}_k} V_k^{(*)}, \Delta \boldsymbol{\theta}_k \rangle + \langle \nabla_{\boldsymbol{x}_k} V_k^{(*)}, \Delta \boldsymbol{x}_k \rangle$. Essentially, minimizing $H(\Delta \boldsymbol{x}_k, \boldsymbol{\theta}_k)$ would minimize the perturbations introduced by any new task.

We simulate worst-case discrepancy by iteratively updating $\Delta \boldsymbol{x}_k$ through gradient ascent, thus maximizing generalization. Next, we minimize forgetting under maximum generalization by iteratively updating $\boldsymbol{\theta}_k$ through gradient descent. To formalize our idea, let us indicate the iteration index at $k$ by $i$ and write $\Delta \boldsymbol{x}_k$ as $\Delta \boldsymbol{x}_k^{(i)}$ and $\boldsymbol{\theta}_k$ as $\boldsymbol{\theta}_k^{(i)}$ with $H(\Delta \boldsymbol{x}_k, \boldsymbol{\theta}_k)$ as $H(\Delta \boldsymbol{x}_k^{(i)}, \boldsymbol{\theta}_k^{(i)})$ (for simplicity of notation, we will denote $H(\Delta \boldsymbol{x}_k^{(i)}, \boldsymbol{\theta}_k^{(i)})$ as $H$ whenever convenient). Towards these updates, we will first get an upper bound on $H(\Delta \boldsymbol{x}_k^{(i)}, \boldsymbol{\theta}_k^{(i)})$ and solve the upper bounding problem.

**Proposition 2.** *Let* $k \in [0, K)$ *and define* $H(\Delta \boldsymbol{x}_k^{(i)}, \boldsymbol{\theta}_k^{(i)}) = \beta_k J_k(\boldsymbol{\theta}_k^{(i)}) + \langle \nabla_{\boldsymbol{\theta}_k} V_k^{(*)}, \Delta \boldsymbol{\theta}_k^{(i)} \rangle + \langle \nabla_{\boldsymbol{x}_k^{(i)}} V_k^{(*)}, \Delta \boldsymbol{x}_k^{(i)} \rangle$ *assume that* $\nabla_{\boldsymbol{\theta}_k} V_k^{(*)} \leq \nabla_{\boldsymbol{\theta}_k} J_k(\boldsymbol{\theta}_k^{(i)})$. *Then the following approximation is true:*

$$
H(\Delta \boldsymbol{x}_k^{(i)}, \boldsymbol{\theta}_k^{(i)}) \leq \beta_k J_k(\boldsymbol{\theta}_k^{(i)}) + (J_k(\boldsymbol{\theta}_k^{(i+\zeta)}) - J_k(\boldsymbol{\theta}_k^{(i)})) + (J_{k+\zeta}(\boldsymbol{\theta}_k^{(i)}) - J_k(\boldsymbol{\theta}_k^{(i)})),
\tag{7}
$$

*where* $\beta_k \in \mathbb{R} \cup [0, 1], \forall k$ *and* $\zeta \in \mathbb{N}$ *and* $J_{k+\zeta}$ *indicates* $\zeta$ *updates on* $\Delta \boldsymbol{x}_k^{(i)}$ *and* $\boldsymbol{\theta}_k^{(i+\zeta)}$ *indicates* $\zeta$ *updates on* $\boldsymbol{\theta}_k^{(i)}$.

*Proof.* Consider $H(\Delta \boldsymbol{x}_k^{(i)}, \boldsymbol{\theta}_k^{(i)}) = \beta_k J_k(\boldsymbol{\theta}_k^{(i)}) + \langle \nabla_{\boldsymbol{\theta}_k} V_k^{(*)}, \Delta \boldsymbol{\theta}_k^{(i)} \rangle + \langle \nabla_{\boldsymbol{x}_k^{(i)}} V_k^{(*)}, \Delta \boldsymbol{x}_k^{(i)} \rangle$. Assuming $\nabla_{\boldsymbol{\theta}_k} V_k^{(*)} \leq \nabla_{\boldsymbol{\theta}_k} J_k(\boldsymbol{\theta}_k^{(i)})$ we may write through finite difference approximation as

$$
\begin{aligned}
\langle \nabla_{\boldsymbol{\theta}_k^{(i)}} V_k^{(*)}, \Delta \boldsymbol{\theta}_k^{(i)} \rangle &\leq \langle \nabla_{\boldsymbol{\theta}_k^{(i)}} J_k(\boldsymbol{\theta}_k^{(i)}), \Delta \boldsymbol{\theta}_k^{(i)} \rangle, \\
&\leq (J_k(\boldsymbol{\theta}_k^{(i+\zeta)}) - J_k(\boldsymbol{\theta}_k^{(i)}))
\end{aligned}
\tag{8}
$$

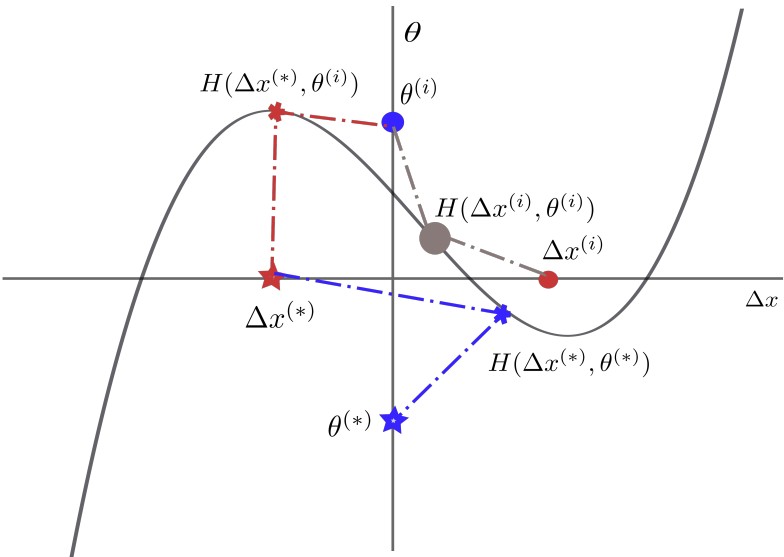

Figure 1: Illustration of proofs. $\Delta x$ (player 1) is the horizontal axis and the vertical axis indicates $\theta$ (player 2) where the curve indicates H. If we start from red circle for player 1 (player 2 is fixed at the blue circle) H is increasing (goes from a grey circle to a red asterisk) with player 1 reaching the red star. Next, start from the blue circle ($\theta$ is at the red star), the cost decreases.

Similarly, we may write

$$
\begin{aligned}
\langle \nabla_{\boldsymbol{x}_k} V_k^{(*)}, \Delta \boldsymbol{x}_k \rangle &\leq \langle \nabla_{\boldsymbol{x}_k} J_k(\boldsymbol{\theta}_k^{(i)}), \Delta \boldsymbol{x}_k \rangle, \\
&\leq (J_{k+\zeta}(\boldsymbol{\theta}_k^{(i)}) - J_k(\boldsymbol{\theta}_k^{(i)})).
\end{aligned}
\tag{9}
$$

Upon substitution, we have our result:

$$
H(\Delta \boldsymbol{x}_k^{(i)}, \boldsymbol{\theta}_k^{(i)}) \leq \beta_k J_k(\boldsymbol{\theta}_k^{(i)}) + (J_k(\boldsymbol{\theta}_k^{(i+\zeta)}) - J_k(\boldsymbol{\theta}_k^{(i)})) + (J_{k+\zeta}(\boldsymbol{\theta}_k^{(i)}) - J_k(\boldsymbol{\theta}_k^{(i)})).
\tag{10}
$$

$\square$

Our cost to be analyzed will be given as

$$
H(\Delta \boldsymbol{x}_k^{(i)}, \boldsymbol{\theta}_k^{(i)}) = \beta_k J_k(\boldsymbol{\theta}_k^{(i)}) + \langle \nabla_{\boldsymbol{\theta}_k^{(i)}} V_k^{(*)}, \Delta \boldsymbol{\theta}_k^{(i)} \rangle + \langle \nabla_{\boldsymbol{x}_k^{(i)}} V_k^{(*)}, \Delta \boldsymbol{x}_k^{(i)} \rangle.
\tag{11}
$$

and use this definition of $H(\Delta \boldsymbol{x}_k^{(i)}, \boldsymbol{\theta}_k^{(i)})$ from here on.

## 2  Main results

We will define two compact sets $\Omega_\theta, \Omega_x$ and seek to show existence and stability of a saddle point $(\Delta \boldsymbol{x}_k^{(i)}, \boldsymbol{\theta}_k^{(i)})$ for a fixed $k$. To illustrate the theory, we refer to Fig. 1, for each $k$, the initial values for the two players are characterized by the pair $\{\boldsymbol{\theta}_k^{(i)}$ (blue circle), $\Delta \boldsymbol{x}_k^{(i)}$ (red circle)$\}$, and $H(\Delta \boldsymbol{x}_k^{(i)}, \boldsymbol{\theta}_k^{(i)})$ is indicated by the grey circle on the cost curve (the dark blue curve). Our proofing strategy is as follows.

First, we fix $\boldsymbol{\theta}_k^{(.)} \in \Omega_\theta$ and construct $\mathcal{M}_k = \{\boldsymbol{\theta}_k^{(.)}, \Omega_x\}$, to prove that H is maximizing with respect to $\Delta \boldsymbol{x}_k^{(i)}$.

**Lemma 2.** *For each* $k \in [0, K)$, *fix* $\boldsymbol{\theta}_k^{(.)} \in \Omega_\theta$ *and construct* $\mathcal{M}_k = \{\Omega_x, \boldsymbol{\theta}_k^{(.)}\}$ *with* $\Omega_\theta, \Omega_x$ *being the sets of all feasible* $\boldsymbol{\theta}_k^{(.)}$ *and* $\boldsymbol{x}_k^{(i)}$ *respectively. Define* $H(\Delta \boldsymbol{x}_k^{(i)}, \boldsymbol{\theta}_k^{(.)})$ *as in Eq. (11) for* $(\Delta \boldsymbol{x}_k^{(i)}, \boldsymbol{\theta}_k^{(.)}) \in \mathcal{M}_k$ *and consider*

$$
\Delta \boldsymbol{x}_k^{(i+1)} - \Delta \boldsymbol{x}_k^{(i)} = \alpha_k^{(i)} \nabla_{\Delta \boldsymbol{x}_k^{(i)}} H(\Delta \boldsymbol{x}_k^{(i)}, \boldsymbol{\theta}_k^{(.)})) / \| \nabla_{\Delta \boldsymbol{x}_k^{(i)}} H(\Delta \boldsymbol{x}_k^{(i)}, \boldsymbol{\theta}_k^{(.)}) \|^2.
$$

*Consider the assumptions* $\nabla_{\boldsymbol{x}_k^{(i)}} V_k^{(*)} \leq \nabla_{\boldsymbol{x}_k^{(i)}} J_k$ *and* $\langle \nabla_{\boldsymbol{x}_k^{(i)}} J_k, \nabla_{\boldsymbol{x}_k^{(i)}} J_k \rangle > 0$, *and let* $\alpha_k^{(i)} \to 0, i \to \infty$. *It follows that* $H(\Delta \boldsymbol{x}_k^{(i)}, \boldsymbol{\theta}_k^{(\cdot)})$ *converges asymptotically to a maximizer.*

*Proof.* Fix $\boldsymbol{\theta}_k^{(\cdot)} \in \Omega_\theta$ and construct $\mathcal{M}_k$ such that $\mathcal{M}_k = \{\boldsymbol{\theta}_k^{(\cdot)}, \Omega_x\}$ which we call a neighborhood. Therefore, for $(\Delta \boldsymbol{x}_k^{(i+1)}, \boldsymbol{\theta}_k^{(\cdot)}), (\Delta \boldsymbol{x}_k^{(i)}, \boldsymbol{\theta}_k^{(\cdot)}) \in \mathcal{M}_k$ we may write a first-order Taylor series expansion of $H(\Delta \boldsymbol{x}_k^{(i+1)}, \boldsymbol{\theta}_k^{(\cdot)})$ around $H(\Delta \boldsymbol{x}_k^{(i)}, \boldsymbol{\theta}_k^{(\cdot)})$ as

$$H(\Delta \boldsymbol{x}_k^{(i+1)}, \boldsymbol{\theta}_k^{(\cdot)}) = H(\Delta \boldsymbol{x}_k^{(i)}, \boldsymbol{\theta}_k^{(\cdot)}) + \langle \nabla_{\Delta \boldsymbol{x}_k^{(i)}} H(\Delta \boldsymbol{x}_k^{(i)}, \boldsymbol{\theta}_k^{(\cdot)}), \Delta \boldsymbol{x}_k^{(i+1)} - \Delta \boldsymbol{x}_k^{(i)} \rangle. \qquad (12)$$

We substitute the update as $\alpha_k^{(i)} \frac{\nabla_{\Delta \boldsymbol{x}_k^{(i)}} H(\Delta \boldsymbol{x}_k^{(i)}, \boldsymbol{\theta}_k^{(\cdot)})}{\|\nabla_{\Delta \boldsymbol{x}_k^{(i)}} H(\Delta \boldsymbol{x}_k^{(i)}, \boldsymbol{\theta}_k^{(\cdot)})\|^2}$ to get

$$H(\Delta \boldsymbol{x}_k^{(i+1)}, \boldsymbol{\theta}_k^{(\cdot)}) - H(\Delta \boldsymbol{x}_k^{(i)}, \boldsymbol{\theta}_k^{(\cdot)}) = \langle \nabla_{\Delta \boldsymbol{x}_k^{(i)}} H(\Delta \boldsymbol{x}_k^{(i)}, \boldsymbol{\theta}_k^{(\cdot)}), \alpha_k^{(i)} \frac{\nabla_{\Delta \boldsymbol{x}_k^{(i)}} H(\Delta \boldsymbol{x}_k^{(i)}, \boldsymbol{\theta}_k^{(\cdot)})}{\|\nabla_{\Delta \boldsymbol{x}_k^{(i)}} H(\Delta \boldsymbol{x}_k^{(i)}, \boldsymbol{\theta}_k^{(\cdot)})\|^2} \rangle. \tag{13}$$

The derivative $\nabla_{\Delta \boldsymbol{x}_k^{(i)}} H(\Delta \boldsymbol{x}_k^{(i)}, \boldsymbol{\theta}_k^{(\cdot)})$ can be written as

$$\begin{aligned} \nabla_{\Delta \boldsymbol{x}_k^{(i)}} H(\Delta \boldsymbol{x}_k^{(i)}, \boldsymbol{\theta}_k^{(\cdot)})) &\leq \nabla_{\Delta \boldsymbol{x}_k^{(i)}} \left[ \beta_k J_k(\boldsymbol{\theta}_k^{(\cdot)}) + \langle \nabla_{\boldsymbol{\theta}_k^{(\cdot)}} V_k^{(*)}, \Delta \boldsymbol{\theta}_k^{(\cdot)} \rangle \right. \\ &\left. + \langle \nabla_{\boldsymbol{x}_k^{(i)}} V_k^{(*)}, \Delta \boldsymbol{x}_k^{(i)} \rangle \right] = \nabla_{\boldsymbol{x}_k^{(i)}} V_k^{(*)} \leq \nabla_{\boldsymbol{x}_k^{(i)}} J_k. \end{aligned} \tag{14}$$

Substitution reveals

$$H(\Delta \boldsymbol{x}_k^{(i+1)}, \boldsymbol{\theta}_k^{(\cdot)}) - H(\Delta \boldsymbol{x}_k^{(i)}, \boldsymbol{\theta}_k^{(\cdot)}) = \alpha_k^{(i)} \frac{\langle \nabla_{\boldsymbol{x}_k^{(i)}} J_k, \nabla_{\boldsymbol{x}_k^{(i)}} J_k \rangle}{\|\nabla_{\boldsymbol{x}_k^{(i)}} J_k\|^2} \tag{15}$$

for $\alpha_k^{(i)} > 0$; and under the assumption that $\langle \nabla_{\boldsymbol{x}_k^{(i)}} J_k, \nabla_{\boldsymbol{x}_k^{(i)}} J_k \rangle > 0$ we obtain

$$H(\Delta \boldsymbol{x}_k^{(i+1)}, \boldsymbol{\theta}_k^{(\cdot)}) - H(\Delta \boldsymbol{x}_k^{(i)}, \boldsymbol{\theta}_k^{(\cdot)}) = \alpha_k^{(i)} \frac{\langle \nabla_{\boldsymbol{x}_k^{(i)}} J_k, \nabla_{\boldsymbol{x}_k^{(i)}} J_k \rangle}{\|\nabla_{\boldsymbol{x}_k^{(i)}} J_k\|^2} \geq 0. \tag{16}$$

Let $B_x = \alpha_k^{(i)} \frac{\langle \nabla_{\boldsymbol{x}_k} J_k, \nabla_{\boldsymbol{x}_k} J_k \rangle}{\|\nabla_{\boldsymbol{x}_k^{(i)}} J_k\|^2} \leq \alpha_k^{(i)}$ and therefore $0 \leq H(\Delta \boldsymbol{x}_k^{(i+1)}, \boldsymbol{\theta}_k^{(\cdot)}) - H(\Delta \boldsymbol{x}_k^{(i)}, \boldsymbol{\theta}_k^{(\cdot)}) \leq \alpha_k^{(i)}$. We therefore have $H(\Delta \boldsymbol{x}_k^{(i+1)}, \boldsymbol{\theta}_k^{(\cdot)}) - H(\Delta \boldsymbol{x}_k^{(i)}, \boldsymbol{\theta}_k^{(\cdot)}) \geq 0$ and $H(\Delta \boldsymbol{x}_k^{(i)}, \boldsymbol{\theta}_k^{(\cdot)})$ is maximizing with respect to $\Delta \boldsymbol{x}_k^{(i)}$. Furthermore, under the assumption that $\alpha_k^{(i)} \to 0, k \to \infty$, we have $H(\Delta \boldsymbol{x}_k^{(i+1)}, \boldsymbol{\theta}_k^{(\cdot)}) - H(\Delta \boldsymbol{x}_k^{(i)}, \boldsymbol{\theta}_k^{(\cdot)}) \to 0$ asymptotically and we have our result. $\qquad\square$

Similarly, we fix $\Delta \boldsymbol{x}_k^{(\cdot)} \in \Omega_x$ and construct $\mathcal{N}_k = \{\Omega_\theta, \Delta \boldsymbol{x}_k^{(\cdot)}\}$, to prove that $H$ is minimizing with respect to $\boldsymbol{\theta}_k^{(i)}$.

**Lemma 3.** *For each $k \in [0, K)$, fix $\Delta \boldsymbol{x}_k^{(\cdot)} \in \Omega_x$ and construct $\mathcal{N}_k = \{\Delta \boldsymbol{x}_k^{(\cdot)}, \Omega_\theta\}$. Then for any $(\Delta \boldsymbol{x}_k^{(i)}, \boldsymbol{\theta}_k^{(\cdot)}) \in \mathcal{N}_k$ define $H(\Delta \boldsymbol{x}_k^{(i)}, \boldsymbol{\theta}_k^{(\cdot)})$ as in Eq. (11) with Proposition. 2 being true and let $\boldsymbol{\theta}_k^{(i+1)} - \boldsymbol{\theta}_k^{(i)} = -\alpha_k^{(i)} \nabla_{\boldsymbol{\theta}_k^{(i)}} H(\Delta \boldsymbol{x}_k^{(\cdot)}, \boldsymbol{\theta}_k^{(i)}))$. Assume that $\|\nabla_{\boldsymbol{\theta}_k^{(i)}} J_k(\boldsymbol{\theta}_k^{(i+\varsigma)})\| \leq L_1$ and $\|\nabla_{\boldsymbol{\theta}_k^{(i)}} J_{k+\varsigma}(\boldsymbol{\theta}_k^{(i)})\| \leq L_2$ and let $\alpha_k^{(i)} \to 0, i \to \infty$. Then $\boldsymbol{\theta}_k^{(i)}$ converges to a local minimizer.*

*Proof.* First, we fix $\Delta \boldsymbol{x}_k^{(\cdot)} \in \Omega_x$ and construct $\mathcal{N}_k = \{\Omega_\theta, \Delta \boldsymbol{x}_k^{(\cdot)}\}$. For any $(\Delta \boldsymbol{x}_k^{(\cdot)}, \boldsymbol{\theta}_k^{(i)}), (\Delta \boldsymbol{x}_k^{(\cdot)}, \boldsymbol{\theta}_k^{(i+1)}) \in \mathcal{N}_k$ we write a first-order Taylor series expansion of $H(\Delta \boldsymbol{x}_k^{(\cdot)}, \boldsymbol{\theta}_k^{(i+1)})$ around $H(\Delta \boldsymbol{x}_k^{(\cdot)}, \boldsymbol{\theta}_k^{(i)})$ to write

$$H(\Delta \boldsymbol{x}_k^{(\cdot)}, \boldsymbol{\theta}_k^{(i+1)}) = H(\Delta \boldsymbol{x}_k^{(\cdot)}, \boldsymbol{\theta}_k^{(i)}) + \langle \nabla_{\boldsymbol{\theta}_k^{(i)}} H(\Delta \boldsymbol{x}_k^{(\cdot)}, \boldsymbol{\theta}_k^{(i)}), \boldsymbol{\theta}_k^{(i+1)} - \boldsymbol{\theta}_k^{(i)} \rangle. \tag{17}$$

We then substitute $\boldsymbol{\theta}_k^{(i+1)} - \boldsymbol{\theta}_k^{(i)} = -\alpha_k^{(i)} \nabla_{\boldsymbol{\theta}_k^{(i)}} H(\Delta\boldsymbol{x}_k^{(.)}, \boldsymbol{\theta}_k^{(i)})$ to get

$$H(\Delta\boldsymbol{x}_k^{(.)}, \boldsymbol{\theta}_k^{(i+1)}) - H(\Delta\boldsymbol{x}_k^{(.)}, \boldsymbol{\theta}_k^{(i)}) = -\alpha_k^{(i)} \langle \nabla_{\boldsymbol{\theta}_k^{(i)}} H(\Delta\boldsymbol{x}_k^{(.)}, \boldsymbol{\theta}_k^{(i)}), \nabla_{\boldsymbol{\theta}_k^{(i)}} H(\Delta\boldsymbol{x}_k^{(.)}, \boldsymbol{\theta}_k^{(i)}) \rangle. \tag{18}$$

Following Proposition 2, the derivative $\nabla_{\boldsymbol{\theta}_k^{(i)}} H(\Delta\boldsymbol{x}_k^{(.)}, \boldsymbol{\theta}_k^{(i)})$ can be written as

$$\nabla_{\boldsymbol{\theta}_k^{(i)}} H(\Delta\boldsymbol{x}_k^{(.)}, \boldsymbol{\theta}_k^{(i)}) \leq \nabla_{\boldsymbol{\theta}_k}[\beta_k J_k(\boldsymbol{\theta}_k^{(i)}) + (J_k(\boldsymbol{\theta}_k^{(i+\zeta)}) - J_k(\boldsymbol{\theta}_k^{(i)})) + (J_{k+\zeta}(\boldsymbol{\theta}_k^{(i)}) - J_k(\boldsymbol{\theta}_k^{(i)}))] \tag{19}$$

Simplification reveals

$$\nabla_{\boldsymbol{\theta}_k^{(i)}} H(\Delta\boldsymbol{x}_k^{(.)}, \boldsymbol{\theta}_k^{(i)})) \leq \nabla_{\boldsymbol{\theta}_k^{(i)}} (\beta_k - 2) J_k(\boldsymbol{\theta}_k^{(i)}) + \nabla_{\boldsymbol{\theta}_k^{(i)}} J_k(\boldsymbol{\theta}_k^{(i+\zeta)}) + \nabla_{\boldsymbol{\theta}_k} J_{k+\zeta}(\boldsymbol{\theta}_k^{(i)}). \tag{20}$$

Substitution therefore provides

$$\begin{aligned}
&H(\Delta\boldsymbol{x}_k^{(.)}, \boldsymbol{\theta}_k^{(i+1)}) - H(\Delta\boldsymbol{x}_k^{(.)}, \boldsymbol{\theta}_k^{(i)}) \\
&\leq -\alpha_k^{(i)} \langle \nabla_{\boldsymbol{\theta}_k^{(i)}} (\beta_k - 2) J_k(\boldsymbol{\theta}_k^{(i)}) + \nabla_{\boldsymbol{\theta}_k^{(i)}} J_k(\boldsymbol{\theta}_k^{(i+\zeta)}) + \nabla_{\boldsymbol{\theta}_k^{(i)}} J_{k+\zeta}(\boldsymbol{\theta}_k^{(i)}), \\
&\quad \nabla_{\boldsymbol{\theta}_k^{(i)}} (\beta_k - 2) J_k(\boldsymbol{\theta}_k^{(i)}) + \nabla_{\boldsymbol{\theta}_k^{(i)}} J_k(\boldsymbol{\theta}_k^{(i+\zeta)}) + \nabla_{\boldsymbol{\theta}_k} J_{k+\zeta}(\boldsymbol{\theta}_k^{(i)}) \rangle.
\end{aligned} \tag{21}$$

Opening the square with Cauchy's inequality provides

$$\begin{aligned}
H(\Delta\boldsymbol{x}_k^{(.)}, \boldsymbol{\theta}_k^{(i+1)}) - H(\Delta\boldsymbol{x}_k^{(.)}, \boldsymbol{\theta}_k^{(i)}) \leq \quad & -\alpha_k^{(i)} \Big[ \|\nabla_{\boldsymbol{\theta}_k^{(i)}} (\beta_k - 2) J_k(\boldsymbol{\theta}_k)\|^2 \\
+ \quad & \|\nabla_{\boldsymbol{\theta}_k^{(i)}} J_k(\boldsymbol{\theta}_k^{(i+\zeta)})\|^2 + \|\nabla_{\boldsymbol{\theta}_k} J_{k+\zeta}(\boldsymbol{\theta}_k^{(i)})\|^2 \\
+ \quad & 2\|\nabla_{\boldsymbol{\theta}_k^{(i)}} (\beta_k - 2) J_k(\boldsymbol{\theta}_k)\| \|\nabla_{\boldsymbol{\theta}_k^{(i)}} J_k(\boldsymbol{\theta}_k^{(i+\zeta)})\| \\
+ \quad & 2\|\nabla_{\boldsymbol{\theta}_k^{(i)}} J_k(\boldsymbol{\theta}_k^{(i+\zeta)})\| \|\nabla_{\boldsymbol{\theta}_k} J_{k+\zeta}(\boldsymbol{\theta}_k^{(i)})\| \\
+ \quad & 2\|\nabla_{\boldsymbol{\theta}_k^{(i)}} (\beta_k - 2) J_k(\boldsymbol{\theta}_k)\| \|\nabla_{\boldsymbol{\theta}_k} J_{k+\zeta}(\boldsymbol{\theta}_k^{(i)})\| \Big].
\end{aligned} \tag{22}$$

We simplify with Young's inequality to achieve

$$\begin{aligned}
H(\Delta\boldsymbol{x}_k^{(.)}, \boldsymbol{\theta}_k^{(i+1)}) - H(\Delta\boldsymbol{x}_k^{(.)}, \boldsymbol{\theta}_k^{(i)}) \leq \quad & -\alpha_k^{(i)} \Big[ \|\nabla_{\boldsymbol{\theta}_k^{(i)}} (\beta_k - 2) J_k(\boldsymbol{\theta}_k)\|^2 \\
+ \quad & \|\nabla_{\boldsymbol{\theta}_k^{(i)}} J_k(\boldsymbol{\theta}_k^{(i+\zeta)})\|^2 \\
+ \quad & \|\nabla_{\boldsymbol{\theta}_k} J_{k+\zeta}(\boldsymbol{\theta}_k^{(i)})\|^2 \\
+ \quad & \|\nabla_{\boldsymbol{\theta}_k^{(i)}} (\beta_k - 2) J_k(\boldsymbol{\theta}_k)\|^2 \\
+ \quad & \|\nabla_{\boldsymbol{\theta}_k^{(i)}} J_k(\boldsymbol{\theta}_k^{(i+\zeta)})\|^2 \\
+ \quad & \|\nabla_{\boldsymbol{\theta}_k^{(i)}} J_k(\boldsymbol{\theta}_k^{(i+\zeta)})\|^2 \\
+ \quad & \|\nabla_{\boldsymbol{\theta}_k} J_{k+\zeta}(\boldsymbol{\theta}_k^{(i)})\|^2 \\
+ \quad & \|\nabla_{\boldsymbol{\theta}_k^{(i)}} (\beta_k - 2) J_k(\boldsymbol{\theta}_k)\|^2 \\
+ \quad & \|\nabla_{\boldsymbol{\theta}_k} J_{k+\zeta}(\boldsymbol{\theta}_k^{(i)})\|^2 \Big].
\end{aligned} \tag{23}$$

Further simplification results in

$$\begin{aligned}
H(\Delta\boldsymbol{x}_k^{(.)}, \boldsymbol{\theta}_k^{(i+1)}) - H(\Delta\boldsymbol{x}_k^{(.)}, \boldsymbol{\theta}_k^{(i)}) \leq \quad & -\alpha_k^{(i)} \Big[ 3\|\nabla_{\boldsymbol{\theta}_k^{(i)}} (\beta_k - 2) J_k(\boldsymbol{\theta}_k)\|^2 \\
+ \quad & 3\|\nabla_{\boldsymbol{\theta}_k^{(i)}} J_k(\boldsymbol{\theta}_k^{(i+\zeta)})\|^2 + 3\|\nabla_{\boldsymbol{\theta}_k} J_{k+\zeta}(\boldsymbol{\theta}_k^{(i)})\|^2 \Big].
\end{aligned} \tag{24}$$

With the assumption that $\|\nabla_{\boldsymbol{\theta}_k^{(i)}} J_k(\boldsymbol{\theta}_k^{(i+\zeta)})\| \leq L_1$ and $\|\nabla_{\boldsymbol{\theta}_k} J_{k+\zeta}(\boldsymbol{\theta}_k^{(i)})\| \leq L_2$, we may write

$$H(\Delta\boldsymbol{x}_k^{(.)}, \boldsymbol{\theta}_k^{(i+1)}) - H(\Delta\boldsymbol{x}_k^{(.)}, \boldsymbol{\theta}_k^{(i)}) \leq \quad -\alpha_k^{(i)} B_\theta, \tag{25}$$

where $B_\theta = \left[((\sqrt{3}\beta_k - 2\sqrt{3})^2 + 3)L_1^2 + 3L_2^2\right]$. Assuming that $\alpha_k^{(i)}$ is chosen such that $\alpha_k^{(i)} \to 0$, we obtain $H(\Delta \boldsymbol{x}_k^{(.)}, \boldsymbol{\theta}_k^{(i+1)}) - H(\Delta \boldsymbol{x}_k^{(.)}, \boldsymbol{\theta}_k^{(i)}) \to 0$ as $i \to \infty$ and $H(\Delta \boldsymbol{x}_k^{(.)}, \boldsymbol{\theta}_k^{(i+1)}) - H(\Delta \boldsymbol{x}_k^{(.)}, \boldsymbol{\theta}_k^{(i)}) < 0$. Therefore H converges to a local minimizer. $\qquad \square$

From here on, we will define our cost function as H whereever convinient for simplicity of notations. Since, for any k, there exists a local maximizer $\Delta \boldsymbol{x}_k^{(*)} \in \Omega_x$, we may define $\mathcal{N}_k^{(*)} = \{\Delta \boldsymbol{x}_k^{(*)}, \Omega_\theta\}$ where the set $\Omega_x$ is comprised of a local maximizer $\Delta \boldsymbol{x}_k^{(*)}$

**Lemma 4.** *For any $k \in [0, K)$, let $\boldsymbol{\theta}_k^{(*)} \in \Omega_\theta$, be the minimizer of H according to Lemma 3 and define $\mathcal{M}_k^{(*)} = \{\Omega_x, \boldsymbol{\theta}_k^{(*)}\}$. Then for $(\Delta \boldsymbol{x}_k^{(*)}, \boldsymbol{\theta}_k^{(*)}), (\Delta \boldsymbol{x}_k^{(i)}, \boldsymbol{\theta}_k^{(*)}) \in \mathcal{M}_k^{(*)}$, $H(\Delta \boldsymbol{x}_k^{(*)}, \boldsymbol{\theta}_k^{(*)}) \geq H(\Delta \boldsymbol{x}_k^{(i)}, \boldsymbol{\theta}_k^{(*)})$, where $\Delta \boldsymbol{x}_k^{(*)}$ is a maximizer for H according to Lemma. 2.*

*Proof.* By Lemma 3, for each $k \in [0, K)$, there exists a minimizer $\boldsymbol{\theta}_k^{(*)} \in \Omega_\theta$ such that $\mathcal{M}_k^{(*)} = \{\Omega_x, \boldsymbol{\theta}_k^{(*)}\}$. Therefore by Lemma 2, $H(\Delta \boldsymbol{x}_k^{(i+1)}, \boldsymbol{\theta}_k^{(*)}) - H(\Delta \boldsymbol{x}_k^{(i)}, \boldsymbol{\theta}_k^{(*)}) \geq 0$ for $(\Delta \boldsymbol{x}_k^{(i+1)}, \boldsymbol{\theta}_k^{(*)}), (\Delta \boldsymbol{x}_k^{(i)}, \boldsymbol{\theta}_k^{(*)}) \in \mathcal{M}_k^{(*)}$. Let $\Delta \boldsymbol{x}_k^{(*)} \in \Omega_x$ be the converging point according to Lemma 2. Then, for $(\Delta \boldsymbol{x}_k^{(*)}, \boldsymbol{\theta}_k^{(*)}), (\Delta \boldsymbol{x}_k^{(i)}, \boldsymbol{\theta}_k^{(*)}) \in \mathcal{M}_k^{(*)}$ a $H(\Delta \boldsymbol{x}_k^{(*)}, \boldsymbol{\theta}_k^{(*)}) - H(\Delta \boldsymbol{x}_k^{(i)}, \boldsymbol{\theta}_k^{(*)}) \geq 0$. by Lemma 2 which provides the result. $\qquad \square$

**Lemma 5.** *For any $k \in [0, K)$, let $\Delta \boldsymbol{x}_k^{(*)} \in \Omega_x$, be the maximizer of H according to Lemma 2 and define $\mathcal{N}_k^{(*)} = \{\Delta \boldsymbol{x}_k^{(*)}, \Omega_\theta\}$. Then for $(\Delta \boldsymbol{x}_k^{(*)}, \boldsymbol{\theta}_k^{(*)}), (\Delta \boldsymbol{x}_k^{(*)}, \boldsymbol{\theta}_k^{(i)}) \in \mathcal{N}_k^{(*)}$, $H(\Delta \boldsymbol{x}_k^{(*)}, \boldsymbol{\theta}_k^{(*)}) \leq H(\Delta \boldsymbol{x}_k^{(*)}, \boldsymbol{\theta}_k^{(i)})$, where $\boldsymbol{\theta}_k^{(*)}$ is a minimizer for H according to Lemma. 2.*

*Proof.* By Lemma 2, for each $k \in [0, K)$, there exists a maximizer $\Delta \boldsymbol{x}_k^{(*)} \in \Omega_x$, such that $\mathcal{N}_k^{(*)} = \{\Delta \boldsymbol{x}_k^{(*)}, \Omega_\theta\}$. Therefore by Lemma 3, $H(\Delta \boldsymbol{x}_k^{(*)}, \boldsymbol{\theta}_k^{(i+1)}) - H(\Delta \boldsymbol{x}_k^{(*)}, \boldsymbol{\theta}_k^{(i)}) \leq 0$ for $(\Delta \boldsymbol{x}_k^{(*)}, \boldsymbol{\theta}_k^{(i+1)}), (\Delta \boldsymbol{x}_k^{(*)}, \boldsymbol{\theta}_k^{(i)}) \in \mathcal{N}_k^{(*)}$. Let $\boldsymbol{\theta}_k^{(*)} \in \Omega_\theta$ be the converging point according to Lemma 3. Then, for $(\Delta \boldsymbol{x}_k^{(*)}, \boldsymbol{\theta}_k^{(*)}), (\Delta \boldsymbol{x}_k^{(*)}, \boldsymbol{\theta}_k^{(i)}) \in \mathcal{M}_k^{(*)}$, $H(\Delta \boldsymbol{x}_k^{(*)}, \boldsymbol{\theta}_k^{(*)}) - H(\Delta \boldsymbol{x}_k^{(*)}, \boldsymbol{\theta}_k^{(i)}) \leq 0$ by Lemma 3 which provides the result. $\qquad \square$

Next, we prove that the union of the two neighborhoods for each k $\mathcal{M}_k^{(*)} \cup \mathcal{N}_k^{(*)}$, is non-empty.

**Lemma 6.** *For any $k \in [0, K)$, let $\boldsymbol{\theta}_k^{(*)} \in \Omega_\theta$, be the minimizer of H according to Lemma 3 and define $\mathcal{M}_k^{(*)} = \{\Omega_x, \boldsymbol{\theta}_k^{(*)}\}$. Similarly, let $\Delta \boldsymbol{x}_k^{(*)} \in \Omega_x$, be the maximizer of H according to Lemma 2 and define $\mathcal{N}_k^{(*)} = \{\Delta \boldsymbol{x}_k^{(*)}, \Omega_\theta\}$. Then, $\mathcal{M}_k^{(*)} \cup \mathcal{N}_k^{(*)}$ is nonempty.*

*Proof.* Let $\mathcal{M}_k^{(*)} \cup \mathcal{N}_k^{(*)}$ be empty. Then, for any $(\Delta \boldsymbol{x}_k^{(i+1)}, \boldsymbol{\theta}_k^{(.)}), (\Delta \boldsymbol{x}_k^{(i)}, \boldsymbol{\theta}_k^{(.)}) \in \mathcal{M}_k^{(*)} \cup \mathcal{N}_k^{(*)}$, $H(\Delta \boldsymbol{x}_k^{(i+1)}, \boldsymbol{\theta}_k^{(.)}) - H(\Delta \boldsymbol{x}_k^{(i)}, \boldsymbol{\theta}_k^{(.)})$ is undefined because the union is empty. This contradicts Lemma 5. Similarly, $H(\Delta \boldsymbol{x}_k^{(.)}, \boldsymbol{\theta}_k^{(i+1)}) - H(\Delta \boldsymbol{x}_k^{(.)}, \boldsymbol{\theta}_k^{(i)})$ for $(\Delta \boldsymbol{x}_k^{(.)}, \boldsymbol{\theta}_k^{(i+1)}), (\Delta \boldsymbol{x}_k^{(.)}, \boldsymbol{\theta}_k^{(i)}) \in \mathcal{M}_k^{(*)} \cup \mathcal{N}_k^{(*)}$ also contradicts Lemma 4. Therefore, by contradiction, $\mathcal{M}_k \cup \mathcal{N}_k$ cannot be empty. $\qquad \square$

## 2.1 Final Results

We are now ready to present the main results. We show that there exists an equilibrium point (Theorem 1 ) and that the equilibrium point is stable (Theorem 2 ).

**Theorem 1** (Existence of an Equilibrium Point)**.** *For any $k \in [0, K)$, let $\boldsymbol{\theta}_k^{(*)} \in \Omega_\theta$, be the minimizer of H according to Lemma 5 and define $\mathcal{M}_k^{(*)} = \{\Omega_x, \boldsymbol{\theta}_k^{(*)}\}$. Similarly, let $\Delta \boldsymbol{x}_k^{(*)} \in \Omega_x$, be the maximizer of H according to Lemma 4 and define $\mathcal{N}_k^{(*)} = \{\Delta \boldsymbol{x}_k^{(*)}, \Omega_\theta\}$. Further, let $\mathcal{M}_k^{(*)} \cup \mathcal{N}_k^{(*)}$ be nonempty according to Lemma. 6, then $(\Delta \boldsymbol{x}_k^{(*)}, \boldsymbol{\theta}_k^{(*)}) \in \mathcal{M}_k^{(*)} \cup \mathcal{N}_k^{(*)}$ is a local equilibrium point.*

*Proof.* By Lemma 5 we have at $(\Delta \boldsymbol{x}_k^{(*)}, \boldsymbol{\theta}_k^{(*)}), (\Delta \boldsymbol{x}_k^{(*)}, \boldsymbol{\theta}_k^{(i)}) \in \mathcal{M}_k^{(*)} \cup \mathcal{N}_k^{(*)}$ that

$$H(\Delta \boldsymbol{x}_k^{(*)}, \boldsymbol{\theta}_k^{(*)}) \leq \quad H(\Delta \boldsymbol{x}_k^{(*)}, \boldsymbol{\theta}_k^{(i)}). \tag{26}$$

Similarly, according to Lemma 4, at $(\Delta \boldsymbol{x}_k^{(*)}, \boldsymbol{\theta}_k^{(*)}), (\Delta \boldsymbol{x}_k^{(i)}, \boldsymbol{\theta}_k^{(*)}) \in \mathcal{M}_k^{(*)} \cup \mathcal{N}_k^{(*)}$ we have

$$H(\Delta \boldsymbol{x}_k^{(*)}, \boldsymbol{\theta}_k^{(*)}) \geq H(\Delta \boldsymbol{x}_k^{(i)}, \boldsymbol{\theta}_k^{(*)}). \tag{27}$$

Putting these inequalities together, we get

$$H(\Delta \boldsymbol{x}_k^{(*)}, \boldsymbol{\theta}_k^{(i)}) \geq H(\Delta \boldsymbol{x}_k^{(*)}, \boldsymbol{\theta}_k^{(*)}) \geq H(\Delta \boldsymbol{x}_k^{(i)}, \boldsymbol{\theta}_k^{(*)}), \tag{28}$$

which is the saddle point condition, and therefore $(\Delta \boldsymbol{x}_k^{(*)}, \boldsymbol{\theta}_k^{(*)})$ is a local equilibrium point in $\mathcal{M}_k^{(*)} \cup \mathcal{N}_k^{(*)}$. $\qquad \square$

According to the preceeding theorem, there is at least one equillibrium point for the game summarized by $H$.

**Theorem 2** (Stability of the Equilibrium Point). *For any* $k \in [0, K)$, $\Delta \boldsymbol{x}_k^{(i)} \in \Omega_x$ *and* $\boldsymbol{\theta}_k^{(i)} \in \Omega_\theta$ *be the initial values for* $\Delta \boldsymbol{x}_k^{(i)}$ *and* $\boldsymbol{\theta}_k^{(i)}$ *respectively. Define* $\mathcal{M}_k = \{\Omega_x, \Omega_\theta\}$ *with* $H(\Delta \boldsymbol{x}_k^{(i)}, \boldsymbol{\theta}_k^{(i)})$ *given by Proposition 2. Let* $\Delta \boldsymbol{x}_k^{(i+1)} - \Delta \boldsymbol{x}_k^{(i)} = \alpha_k^{(i)} \times (\nabla_{\Delta \boldsymbol{x}_k^{(i)}} H(\Delta \boldsymbol{x}_k^{(i)}, \boldsymbol{\theta}_k^{(\cdot)}))/\|\nabla_{\Delta \boldsymbol{x}_k^{(i)}} H(\Delta \boldsymbol{x}_k^{(i)}, \boldsymbol{\theta}_k^{(\cdot)})\|^2)$ *and* $\boldsymbol{\theta}_k^{(i+1)} - \boldsymbol{\theta}_k^{(i)} = -\alpha_k^{(i)} \times \nabla_{\boldsymbol{\theta}_k^{(i)}} H(\Delta \boldsymbol{x}_k^{(\cdot)}, \boldsymbol{\theta}_k^{(i)})$. *Let the existence of an equilibrium point be given by Theorem 1, then as a consequence of Lemma 2 and 3* $(\Delta \boldsymbol{x}_k^{(*)}, \boldsymbol{\theta}_k^{(*)}) \in \mathcal{M}_k$ *is a stable equilibrium point for* $H$.

*Proof.* Consider now the order of plays by the two players. By Lemma 2, a game starting at $(\Delta \boldsymbol{x}_k^{(i)}, \boldsymbol{\theta}_k^{(i)}) \in \mathcal{M}_k$ will reach $(\Delta \boldsymbol{x}_k^{(*)}, \boldsymbol{\theta}_k^{(i)})$ which is a maximizer for $H$. Now, define $\mathcal{N}_k = (\Delta \boldsymbol{x}_k^{(*)}, \Omega_\theta) \subset \mathcal{M}_k$ then a game starting at $(\Delta \boldsymbol{x}_k^{(*)}, \boldsymbol{\theta}_k^{(i)}) \in \mathcal{N}_k$ will converge to $(\Delta \boldsymbol{x}_k^{(*)}, \boldsymbol{\theta}_k^{(*)}) \in \mathcal{N}_k$ according to Lemma 3. Since, $\mathcal{N}_k \subset \mathcal{M}_k$, our result follows. $\qquad \square$

## 3   Experimental Details

Much of this information is a repetition of details provided in [3, 11].

1. *Incremental Domain Learning  (IDL ):* Incremental domain refers to the scenario when each new task introduces changes in the marginal distribution of the inputs. This scenario has been extensively discussed in the domain adaptation literature, where this shift in domain is typically referred to as "non-stationary data distribution" or domain shift. Overall, we aim to transfer knowledge from the old task to a new task where each task can be different in the sense of their marginal distribution.

2. *Incremental Class Learning  (ICL ):* In this scenario, each task contains an exclusive subset of classes. The number of output nodes in a model equals the number of total classes in the task sequence. For instance, tasks could be constructed by using exactly one class from the MNIST data set where we aim to transfer knowledge from one class to another.

3. *Incremental Task Learning  (ITL ):* In this setup, the output spaces are disjoint between tasks[ for example, the previous task can be a classification problem of five classes, while the new task can be a regression. This scenario is the most generic and allows for the tasks to be defined arbitrarily. For each tasks, a model requires task-specific identifier$t$.

**Split-MNIST** For split-MNIST, the original MNIST-data set is split into five partitions where each partition is a two-way classification. We pick 60000 images for training (6000 per digit ) and 10000 images for test, i.e. (1000 per digit ). For the incremental task learning in the split-MNIST experiment, the ten digits are split into five two-class classification tasks (the model has five output heads, one for each task ) and the task identity (1 to 5 ) is given for test. For the incremental class learning setup, we require the model to make a prediction over all classes (digits 0 to 9 ). For the incremental domain learning, the model always predicts over two classes.

**Permuted-MNIST** For permuted-MNIST, we permute the pixels in the MNIST data to create tasks where each task is a ten-way classification. The three CL scenarios that are generated for the permuted-MNIST are similar to the Split-MNIST data set except for the idea that the different tasks are now generated by applying random pixel permutations to the images. For incremental task learning, we use a multi-output strategy, and each task is attached to a task identifier. For incremental domain and class, we use a single output strategy and each task as one where one of the 10 digits are predicted. In incremental class learning, for each new task 10 new classes are generated by permuting the MNIST data set. For incremental task and domain, we use a total of 10 tasks whereas for incremental classes, we generate a total of 100 tasks.

**Network Architecture** We keep our architecture identical to what is provided in [3, 11]. The loss function is categorical cross-entropy for classification. All models were trained for 2 epochs per task with a minibatch size of 128 using the Adam optimizer ($\beta_1 = 0.9$, $\beta_2 = 0.999$, learning rate= 0.001 ) as the default. For BCL, the size of the buffer (i.e., a new task array ) $\mathcal{D}_N(k)$ and a task memory array (samples from all the previous tasks ) $\mathcal{D}_P(k)$) is kept equivalent to naive rehearsal and other memory-driven approaches such as GEM and MER (16, 000 samples ).

**Comparison Methods – Baseline Strategies** Additional details can be found from [3, 11]

1. A sequentially-trained neural network with different optimizers such as SGD, Adam [4], and Adagrad [2].
2. A standard $L_2-$regularized neural network where each task is shown sequentially.
3. Naive rehearsal strategy (experience replay ) where a replay buffer is initialized and data corresponding to all the previous tasks are stored. The buffer size is chosen to match the space overhead of online EWC and SI.

**Comparison Method-CL** We compared the following CL methods:

1. **EWC [5] / Online EWC [9] / SI [12]**: For these methods, a regularization term is added to the loss, with a hyperparameter used to control the regularization strength such that: $L(total) = L(current) + \lambda L(regularization)$. $\lambda$ is set through a hyperparameter.
2. **LwF [6] / DGR [10]** Here, we set the loss to be $L(total) = \alpha L(current) + (1\alpha)L(replay)$ where hyperparameter $\alpha$ is chosen according to how many tasks have been seen by the model.
3. For **RtF** [11], MAS [1], GEM[7] and MER[8], we refer to the respective publication for details.

Additional details about the experiments can be found in [3] as our paper retains their hyper-parameters and the experimental settings.