# OpenReview forum: "Formalizing the Generalization-Forgetting Trade-off in Continual Learning"
_NeurIPS.cc/2021/Conference — NeurIPS 2021 Poster_

### Official Review · Reviewer_jBBb · 2021-07-16

**Rating:** 7
**Confidence:** 5

**Summary:**

The paper attempts to present a theoretical framework for modelling the trade-off between generalization (adaptation to future tasks, plasticity) and forgetting (stability on previous tasks). Towards this, the paper proposes an objective function that minimizes the weighted loss on all the past (implemented using episodic memory), present (implemented using the dataset of the current task) and future tasks (approximated using preturbing the datasets of the past and current tasks). The perturbation to this loss is then approximated in the local neighbourhood of the current task. The optimization of the loss is posed as a minimax game between the two players, where player 1 maximizes the generalization by maximizing the discrepancy between the successive tasks, and player 2 minimizes the forgetting by adapting the model parameters against that discrepancy. The authors show that if player 1 starts first, the equilibrium solution of this game exists in the local neighbourhood that is defined using the current task. The proposed approach achieves strong results against several baselines on MNIST and CIFAR-100 variants used for continual learning.


**Limitations And Societal Impact:**

Yes.

**Main Review:**

Positives:
- The paper is written very well and the generalization-forgetting trade-off is well-motivated.
- There is a dearth of theory papers in continual learning and this presents a very nice theory for analyzing the trade-off between forgetting and generalization.
- Empirical results are quite strong.

Negatives:

While I don’t have many problems with the paper, I do like to ask a couple of questions.

- Modelling generalization: The paper argues that the degree of generalization depends on the discrepancy between the previous and new tasks. Hence in maximizing that discrepancy, and subsequently training your model to account for that discrepancy, we maximize the generalization. I could see why as an optimization objective such an adversarial training would encourage the model to adapt to potentially different tasks. But you can’t say that a model would generalize better to more different tasks, in fact it is the opposite. If the two tasks are similar then the model has a better chance of generalizing (it's a test time concept). Maybe the authors would want to rephrase the lines 121-125 to correctly reflect this.

- Time constraints: It seems that the proposed approach BCL is quite time consuming. Could the authors please provide the timing comparison of their approach with the other baselines?


**Time Spent Reviewing:**

4

---

> ### Author Response · Authors · 2021-08-10
> **Response for Reviewer jBBb**
>
> Response (About modelling generalization)   As the reviewer suggested, we will clarify this in the revised version.  We agree that a large difference between tasks would provide detrimental effects on the performance of the model.  In fact, we claim that the more similar the tasks, the better the performance of the model. This idea surfaces itself in the proof of convergence (Lemma 2, 3) where we must assume boundedness on $\Delta x$ for the proofs’ to work. Boundedness of $\Delta x$ indicates that the new task must not be too different from the old task. This is therefore a limitation for any CL approach (unless one learns a transformation between tasks, even then, the limitation would come from the efficiency of the transport mechanism between task distributions).
> Furthermore, the idea of generalization in this context is the ability to adapt to new tasks during training that is distinct from the train to test time generalization concept.
>
>
> Response ( Time constraints): A calculation of computational complexity is as follows:  for every batch of data, we have to first evaluate the cost function, then perform a gradient update using the cost H. To evaluate H, we perform 2 loops of zeta steps each (refer to the algorithm in the main paper). Consider the complexity of calculating and updating parameters in a standard NN using one batch of data be denoted as $O(g)$. Then the complexity of the two inner loops is $2 \zeta O[g]$.  Next, we use $H$ to update the parameters of the network once which lets say has the complexity $O(g)$. Therefore the rough computational complexity of our method is $O(g) (1+2 \zeta ),$ where $O(g)$ is the complexity of one gradient calculation and update.
> For comparison, MER comprises two cascaded loops with zeta steps each.  The complexity of inner loop would be $O[g] * \zeta$ and the complexity of outer loop would be $\zeta* O[R]$, where $O[R]$ is the complexity of reptile update. Thus rough calculation will provide the complexity as $\zeta^2 \times O[R] \times O[g].$ MER is quadratic with respect to $\zeta$ whereas ours is linear with respect to $\zeta.$  Furthermore, the complexity of MAS and GEM is similar to that of MER and thus $\zeta^2 \times O[R] \times O[g].$  Other generative replay methods such as RtF are linear with respect to $O[g].$  The complexity of EWC can be ensured to be $p*O[g]$ (with simplifying assumptions), where p is the number of parameters in the network.

---

### Official Review · Reviewer_dfQR · 2021-07-16

**Rating:** 7
**Confidence:** 4

**Summary:**

The paper proposed a novel idea of formalizing the generality-forgetting trade-off in continual learning via a two-player sequential game. Theoretical and empirical results are represented to support the authors' claims. The main contribution of the paper is to formally discuss the forgetting and generalization trade-off theoretically.

**Limitations And Societal Impact:**

Yes.

**Main Review:**

The idea of formalizing continual learning problem as a two-player game is novel and interesting. The theorems are interesting, but the flow of the paper is not quite clear, which makes the details hard to fully understand. The paper has put a lot of emphasis on its theoretical contribution, however, it is not quite clear in their experiments. Showing the average prediction accuracy in benchmark datasets, although necessary, could not fully support the theoretical side. When talking about trade-off theoretically, authors should think about ways to reveal it experimentally also.  For example, several metrics such as average forgetting and learning curve area [1] could be leveraged to depict the actual trade-off between forgetting and generalization. Also, in Table 2, showing more recent methods instead of earlier methods that are not particularly designed for incremental class learning might be a better choice.


[1] EFFICIENT LIFELONG LEARNING WITH A-GEM, Chaudhry et al.

**Time Spent Reviewing:**

4

---

> ### Author Response · Authors · 2021-08-10
> **Response for Reviewer dfQR**
>
> The main element of our theory is to show that there exists a balance point between generalization and forgetting. Furthermore, we argue that modelling this balance point improves performance. One way we attempted to show the effect of this modelling is to show that when we stop modelling the tradeoff and switch off the game, we observe a deterioration in performance for our method. These results are summarized in the last two columns of Table 1 and 2. In addition to this, we will also include a plot of average forgetting and generalization which reveals this trade-off in the revision.
>
> For our experimental section we rely on the continual learning benchmark [1] and include relevant methods that model the trade-off such as MER[2] (the only method that explicitly models the trade-off) in it.
>
> [1] Yen-Chang Hsu, Yen-Cheng Liu, Anita Ramasamy, and Zsolt Kira.  Re-evaluating continual learning scenarios:  A categorization and case for strong baselines.   In NeurIPS Continual learning Workshop , 2018.
>
> [2] Matthew Riemer, Ignacio Cases, Robert Ajemian, Miao Liu, Irina Rish, Yuhai Tu, and Gerald Tesauro.   Learning to learn without forgetting by maximizing transfer and minimizing interference. arXiv preprint arXiv:1810.11910, 2018

---

> > ### Comment · Reviewer_dfQR · 2021-08-31
> > **Follow-up**
> >
> > I thank the authors for their clarification and explanation. After double-checking the points, I realized the theoretical highlights of the paper, although I still think the way of presenting the ideas should be more intuitive. I am happy to raise my score to acceptance.

---

> ### Author Response · Authors · 2021-08-21
> **Clarification regarding the two main concerns: unclear presentation and lack of comprehensive experiments**
>
> Regarding the first concern, as pointed out by other reviewers (specifically, Reviewer Me2K and Reviewer jBBb), the paper is in fact clear in its presentation.
>
> Regarding the second concern:  We use the continual learning benchmark published in NeurIPS’ 18  continual learning workshop. Within this framework, we compared a total of 14 methods over a total of 9 dataset instances (both MNIST and CIFAR-100 are used with split and incremental strategies to create these 9 dataset instances). We used three scenarios: incremental class (task independent), incremental domain (task independent) and incremental task (task dependent) scenarios. Experimental results of our method on these  scenarios indicate varied and diverse applicability for our method. For the 6 instances of MNIST dataset we repeat the $(14 \times 6)$ experiments 20 times and publish the mean and variance of the retained accuracy (metric utilized by both [1] and [2]). For the three instances of CIFAR100, we copy the numbers for other methods from [1] and run 20 repetitions for BCL. Prior to our work, the only method that attempts to heuristically and explicitly balance generalization and forgetting is Meta Experience Replay~(MER) [2]. We implement MER within the continual learning benchmark and compare them to our BCL method. The code is made available for all of these methods and we demonstrate considerable performance improvement achieved by BCL in all these experiments.
>
> Recent papers in continual learning literature (published in NeurIPS, 2019 and 2020) utilize data-sets and methods similar to our paper. For example, LAMAML--look ahead model agnostic meta learning, Gupta et. al, NeurIPS’20 [3] use cifar100 and report retained accuracy (same metric as our paper) with comparisons to variants of ER (experience reply), GEM (gradient episodic memory) and MER. Similarly, MCL by Javed et.al, NeurIPS’19 [4] compared to experience replay, MER and GEM. We have utilized all these methods in our paper too. Moreover, our experimental setting is in fact much more comprehensive than these methods as we consider three different continual learning scenarios (incremental class, incremental task and incremental domain) whereas both Gupta’ et al and  Javed et. al consider only the incremental class setting.
>
> The call for NeurIPS'21,  specifically invites original research in theory  (eighth bullet point, i.e, Theory (e.g., control theory, learning theory, algorithmic game theory)). Our paper is on  development and application of game theory to continual learning with novel theoretical results in the field of continual learning. All the reviewers have commented that the work is novel and the theoretical results are significant. Even with the considerable experiments conducted as part of this paper that are at par with the recent papers published in NeurIPS.
>
> [1] Yen-Chang Hsu, Yen-Cheng Liu, Anita Ramasamy, and Zsolt Kira.  Re-evaluating continual learning scenarios:  A categorization and case for strong baselines.   In NeurIPS Continual learning Workshop , 2018.
> [2] Riemer, Matthew, et al. "Learning to learn without forgetting by maximizing transfer and minimizing interference." arXiv preprint arXiv:1810.11910 (2018).
> [3] Gupta, Gunshi, Karmesh Yadav, and Liam Paull. "La-maml: Look-ahead meta learning for continual learning." arXiv preprint arXiv:2007.13904 (2020).
> [4] Javed, Khurram, and Martha White. "Meta-learning representations for continual learning." arXiv preprint arXiv:1905.12588 (2019).

---

### Official Review · Reviewer_Me2K · 2021-07-16

**Rating:** 7
**Confidence:** 4

**Summary:**

The paper theoretically investigates the trade-off between generalization and forgetting in continual learning (CL). The authors prove the existance of a saddle point in the generalization/forgetting space; they show that it is stable under certain conditions; and propose a practical algorithm called balanced continual learning (BCL) for finding it. Finally, the authors compare their approach experimentally against the state of the art in CL.

**Limitations And Societal Impact:**

Yes

**Main Review:**

In my opinion, the paper is original and contains some very interesting ideas (i.e., the generalization/forgetting trade-off and using gradient ascent to improve generalization).

The text is mostly clear, but I think the authors should try to simplify their mathematical notation, as it is very dense and very difficult to wrap your head around it.

The theoretical results and the algorithm presented in the paper are significant for non-task-free CL settings.

I feel that the experiments of the paper are insufficient. Two-thirds of the experimental work are based on the MNIST dataset which is extremely simplistic and cannot be considered to give trustworthy benchmarks. The Permuted MNIST benchmark, in particular, has been (in my opinion, justly) criticized by other works (e.g., Farquhar and Gal, 2018) for not being realistic enough. I recommend that the authors include at least one or two more split-type benchmarks in their experimental work (e.g., CIFAR-10 and/or mini-ImageNet). I would also recommend studying the effect of memory size on the behavior of the proposed algorithm using one of the 'harder' datasets.

Finally, I think that the paper will benefit a lot by if the authors were to describe some intuitive interpretations of the main theoretical results of their work.


Minor Issues:

- Line 96: There is a "Third" left there (probably after re-writing the paragraph) that needs to be removed.
- Line 134: In line 3 of Eq. (3), insert a large enough gap between the min and max operators so that their corresponding subscripts are clearly seperated.
- Line 210: Regarding the expectation operator, you have used an italic E in the formula but a regular E in the text. In any case, I would recommend using a blackboard bold typeface for the expectation operator.


Questions:

- What is the effect of memory size on the BCL game?
- Do you think BCL can be expanded for task-free CL?
- Do we really need to prioritize tasks? Why not use the expectation of all task losses (which will be bounded given the conditions stated in the paper)?
- Why define the loss function to be a weighted sum of H, where H is a weighted sum of task losses? Why not just use a weighted sum of task losses directly?
- What is the intuition behind the selected order of play (Δx first, θ afterwards)?
- Have you experimentally tried using the reverse order of play? I am curious if the results would be different as Theorem 2 only discusses the selected order of play but not the reverse.

**Time Spent Reviewing:**

4

---

> ### Author Response · Authors · 2021-08-10
> **Response for Reviewer Me2K**
>
> Response (Mathematical Notations):In the revision we will simplify the notations and move additional complexity to the appendix.
>
> Response (additional experiments):We have included CIFAR-100 results with split strategies in the paper (refer Table 2) that prove the superior performance of balanced continual learning.
>
> Response (Intuitive explanations): In addition to the intuitive explanations present in the paper (refer to section 3.1, specifically under the theorem statements, where we explain our proofing strategy intuitively), we will include additional explanations in the revision. Specifically,
> We will further expand the explanations on line number 82-92 to clarify the impact of the cost formulation.
> We will clarify the intuitions behind Eq. 2 in lines 106-120.
> We will expand the explanation in section 3.1 to clarify the implications of our theory.
> We will add explanations in the results to demonstrate connections between the theory and the results.
>
>
> Response (typographical errors): We will fix these in the revision.
>
> Questions:
> What is the effect of memory size on the BCL game?
>
> Response: A larger memory size would allow us to store  more examples (more data corresponding to tasks) for BCL to utilize. However, the choice of memory is proportional to the tasks, for example, for a small number of tasks, a small memory size could be sufficient to store representative examples. On the other hand, for a large number of tasks, a small memory size may not be sufficient. In such a scenario, according to Lemma 1 and Corollary 1, task prioritization is crucial.
>
> Do you think BCL can be expanded for task-free CL?
>
> Response: Yes, our theoretical framework can be expanded to task-free CL. Refer to Algorithm 1, where we do not require any task identifiers for learning.  Furthermore, in the experiments, we implemented our method in incremental task setting (where task identifier is required) and incremental class and domain setting (where task identifier is not required).
>
> Do we really need to prioritize tasks? Why not use the expectation of all task losses (which will be bounded given the conditions stated in the paper)?
>
> Response: Prioritization of tasks is not mandatory and becomes important only when a large number of tasks are involved (refer lines 82-92 and note lemma 1 and corollary 1 are proven in the limit (number of tasks go to infinity)). When a large number of tasks are present, the lack of priority will introduce large forgetting on the previous tasks. No matter how much the game tries to find a balance point, when the number of tasks are large, the game will tend to lose performance on previous tasks as proven in Corollary 1. The only way to guide the game towards particular tasks is to prioritize. This is a drawback of any methodology that tries to achieve balance.
>
> Why define the loss function to be a weighted sum of H, where H is a weighted sum of task losses? Why not just use a weighted sum of task losses directly?
>
> Response: H is derived as part of the framework and therefore equal to all the values within the bracket in Eq. 2. Since, the theoretical value must be approximated, we derive this approximation in proposition 2.  H is in turn derived from the original formulation in Eq. 1.
>
> What is the intuition behind the selected order of play (Δx first, θ afterwards)?  Have you experimentally tried using the reverse order of play? I am curious if the results would be different as Theorem 2 only discusses the selected order of play but not the reverse.
>
> Response: Δx  refers to the change in subsequent tasks. Therefore the value of Δx  is dictated by the data, is unknown and is beyond the control of the designer. However, our goal is to adapt to Δx~(the model cannot adapt to what is unknown about the tasks). Therefore, we simulate the unknown change by maximizing the impact of Δx and minimize the impact of such change using θ. This is the intuition behind the order of play.
> Moreover, the order of play is important. Only when subsequent tasks are different from the older tasks,  Δx is non-zero. Furthermore, only when such a difference is detected by the model (through the value of the loss), the model can adapt to the change in the tasks. Therefore, it is necessary for the tasks to change and provide a nonzero  Δx before the model can adapt to it. Reversing the order of play therefore does not coincide with the objectives of continual learning (the problem).  This intuition is described in lines 121-134 and we will clarify it further in the revision.

---

> > ### Comment · Reviewer_Me2K · 2021-09-03
> > **Response to Rebuttal**
> >
> > I'd like to thank the authors for their rebuttal. The majority of my concerns were resolved so I decided to increase my score accordingly.

---

### Official Review · Reviewer_aHYF · 2021-07-18

**Rating:** 7
**Confidence:** 3

**Summary:**

This paper considers the stability-plasticity dilemma in continual learning. The authors develop an algorithm which addresses catastrophic forgetting while trying to improve generalisation on new tasks.

The paper introduces a new cost function for continual learning, which takes into account the model's performance on previous, current and future tasks. An approximation, H, to this cost function, which involves only the data that has been encountered so far, is optimised. The optimisation is phrased as a 2-player game, where player 1 tries to maximise H by creating worst-case dissimilar inputs for future tasks. Player 2 minimises the H by optimising the model's parameters. The authors show that the two players would eventially reach a single solution (an equilibrium).
Catastrophic forgetting is prevented by maintaining a subset of previously encountered datasets and using it when optimising H.

Contributions:
- A novel formulation of the cost function of continual learning (CL)
- A theoretically justified approach to CL.
- An implementation of their approach
- Experiments which show that the resulting method outperforms the many baselines.


**Limitations And Societal Impact:**

Yes.

**Main Review:**

Disclaimer: I did not verify the proofs in the appendix.

It was interesting for me to read the new CL cost function. The analysis of the cost function and the proposed 2-player approach was also informative.
The experiments evaluate different CL settings (class-incremental, domain-incremental and task-incremental), which I think is useful for determining the overall worth of a proposed new method.


Weaknesses:
[Experiments] I'd be curious to see experiments with mixed datasets (e.g. CIFAR10 and MNIST), which would better illustrate whether your method can generate to diverse input domains

Note: I'm not sure that the baselines used in the experiments are state-of-the-art.

Note: [Clarity] It appears that Algorithm 1 has j=0 and i=0 swapped. Needs to be corrected.

Questions regarding the cost function H on line 206:
q1: When approximating H for player 2, you copy over the parameters in a new network \theta_B and optimise them. Afterwards, you use the predictions of this optimised network to calculate H. However, these predictions are not differentiable wrt the original \theta , are they? If this is the case, why calculate it at all?
q2: H includes J_k(\theta) a number of times: H = \beta * J_k(\theta) + ... - J_k(\theta) + ... -J_k(\theta) = J_k(\theta) *(\beta - 2) + ... .
If \beta < 2,  when player 2 is minising H, wouldn't it try to increase J_k(\theta) in order to minimise H? In this case, I find it a bit confusing why the method works so well?
I looked at your code as well to try to find the answers to the above questions, but it's not clear to me. Looking at the code, it appears that you are using \beta = 1.

q3: How computationally intensive would you say your method is, compared to others?


**Time Spent Reviewing:**

8

---

> ### Author Response · Authors · 2021-08-10
> **Response for Reviewer  aHYF**
>
>
> Response for comment on experimental results :  Our experimental comparisons in Tables 1 and 2, comprise two classes of methods: baseline methods and continual learning methods. In baseline experiments, we use Adam, experience replay (Naive rehearsal and naive rehearsal-C) and $L_2$. For other experiments, we focus on traditional continual learning methods such as elastic weight consolidation~(EWC),  synaptic intelligence(SI) and newer continual learning methods such as memory aware synapses~(MAS), meta experience replay~(MER), deep generative replay~(DGR) and generative replay with feedback connections~(RtF). Since, our method seeks to achieve balance between forgetting and generalization, we compare the only method in recent years to model this balance, that is, MER. To further substantiate our results, we will add an experiment with mixed datasets (where tasks are distinct with each other, e.g. Mnist and cifar10) in the revision.
>
>
> Response(Note 1) This will be corrected in the revision.
>
> Response (Q1): As described in line numbers (227-230), the use of $\theta_B$ is to provide a value for the term $J_k(\theta^{i+\zeta})$  in $J_k(\theta^{i+\zeta})-J_k(\theta^{i}).$ In other words, we replace the value of $\theta^{i+\zeta}_{k}$ by $\theta_B$~(after $\zeta$ updates). Therefore, $J_k(\theta_B)$  acts a target for $J_k(\theta^{i})$ in $J_k(\theta_B)-J_k(\theta^{i})$ and it is not necessary for the gradient to flow through $\theta_B.$ Note that $J_k(\theta^{i})$ is still calculated using the original weights.
>
> Response(Q2):
> Since this is a two player game, the balance is achieved only when terms in the cost cancel each other. Therefore, even though some terms behave opposite to what is expected out of them, the overall cost will still go to zero.  We have added the following explanation to the revised version of the paper.
> Intuitively, we care about the collective behavior of H instead of individual terms in the cost. For example, take $\beta=1,$ Then you will get $H= -J_k(\theta) + J_{k+\zeta}(\theta) + J_{k+\zeta}(\theta^{i+\zeta})$. Now, as long as  $J_{k+\zeta}(\theta) + J_{k+\zeta}(\theta^{i+\zeta}) > J_k(\theta),$ then H will keep going down. In fact, the game will make $H$ go up sometimes and down sometime, that is, H will oscillate. The balance point is achieved when $J_{k+\zeta}(\theta) + J_{k+\zeta}(\theta^{i+\zeta}) =  J_k(\theta).$
> To observe this behavior theoretically, note the proof of Lemma~3 (line 116-118 in supplementary), where $\beta-2$ comes in the proof. Although what you observed is correct for that individual term, the overall norm operator that acts on $\beta-2$ makes it positive and we get the behavior we want. We will add this explanation to section 3.2 of the paper.
>
>
> Response (Q3): A calculation of computational complexity is as follows:  for every batch of data, we have to first evaluate the cost function, then perform a gradient update using the cost H. To evaluate H, we perform 2 loops of zeta steps each (refer to the algorithm in the main paper). Consider the complexity of calculating and updating parameters in a standard NN using one batch of data be denoted as $O(g)$. Then the complexity of the two inner loops is $2 \zeta O[g]$.  Next, we use $H$ to update the parameters of the network once which lets say has the complexity $O[g]$. Therefore the rough computational complexity of our method is $O(g) (1+2 \zeta ),$ where $O[g]$ is the complexity of one gradient calculation and update.
> For comparison, MER comprises two cascaded loops with zeta steps each.  The complexity of inner loop would be $O[g] \zeta$ and the complexity of outer loop would be $\zeta  O[R]$, where $O[R]$ is the complexity of reptile update. Thus rough calculation will provide the complexity as $\zeta^2 O[R]  O[g].$ MER is quadratic with respect to $\zeta$ whereas ours is linear with respect to $\zeta.$  Furthermore, the complexity of MAS and GEM is similar to that of MER and thus $\zeta^2 \times O[R] \times O[g].$  Other generative replay methods such as RtF are linear with respect to $O[g].$  The complexity of EWC can be ensured to be $p O[g]$ (with simplifying assumptions), where $p$ is the number of parameters in the network.
>
> We will add this explanation on complexity to the paper in the revision.

---

### Author Response · Authors · 2021-08-10
**We thank the reviewers for their valuable feedback. The main focus of our paper is theoretical and in this pursuit we have developed methods for studying the CL problem and its challenges in depth.**

As observed by the reviewers, our contributions are as follows.

A novel cost formulation  for continual learning (CL)
Balanced continual learning (BCL): A theoretically justified approach to CL.
Demonstration of BCL on open source CL benchmarks and comparison to 14 different CL methods.
Experiments that show that the resulting method outperforms several previously proposed methods.

The main focus of our paper is theoretical and in this pursuit we have developed methods for studying the CL problem and its challenges in depth.

For our experiments, we used the continual learning benchmark published in NeurIPS’ 18  continual learning workshop [1] (cited 108 times according to Google scholar). Within this framework, we compared a total of 14 methods over a total of 9 dataset instances (both MNIST and CIFAR-100 are used with split and incremental strategies to create these 9 dataset instances). We used three scenarios: incremental class (task independent), incremental domain (task independent) and incremental task (task dependent) scenarios. Experimental results of our method on these  scenarios indicate varied and diverse applicability for our method. For the 6 instances of MNIST dataset we repeated the $(14 \times 6)$ experiments 20 times and publish the mean and variance of the retained accuracy (metric utilized by both [1] and [2]). For the three instances of CIFAR100, we copied the numbers for other methods from [1] and run 20 repetitions for BCL. Prior to our work, the only method that attempts to heuristically and explicitly balance generalization and forgetting is Meta Experience Replay~(MER) [2]. We implement MER within the continual learning benchmark and compare them to our BCL method. The code is made available for all of these methods and we demonstrate considerable performance improvement achieved  by BCL in all these experiments.


[1] Yen-Chang Hsu, Yen-Cheng Liu, Anita Ramasamy, and Zsolt Kira.  Re-evaluating continual learning scenarios:  A categorization and case for strong baselines.   In NeurIPS Continual learning Workshop , 2018.

[2] Matthew Riemer, Ignacio Cases, Robert Ajemian, Miao Liu, Irina Rish, Yuhai Tu, and Gerald Tesauro.   Learning to learn without forgetting by maximizing transfer and minimizing interference. arXiv preprint arXiv:1810.11910, 2018

---

### Decision · Program_Chairs · 2021-09-27

**Decision:**

Accept (Poster)

**Comment:**

The paper addresses the balance between stability-plasticity tradeoff as a two player sequential game and​​ show theoretically that a balance point between the two players exists for each task and that this point is stable. This leads to a theoretically justified approach, balanced continual learning (BCL), which has empirical performance that is comparable to or better than the state of the art CL approaches.

Overall, the paper addresses an important question, provides a new perspective of formulating the cost, provides theoretical justification as well as empirical evaluation for the proposed method. The reviewers raised some points that were satisfactorily addressed in the rebuttal, which helped improved their understanding of where the paper stands.